# Early-Exit Neural Networks with Nested Prediction Sets

**Metod Jazbec**[*1,2]    **Patrick Forré**[2]    **Stephan Mandt**[3]    **Dan Zhang**[4]    **Eric Nalisnick**[1,2]

[1] UvA-Bosch Delta Lab, University of Amsterdam
[2] Amsterdam Machine Learning Lab, University of Amsterdam
[3] Department of Computer Science, University of California, Irvine
[4] Bosch Center for AI & University of Tübingen

## Abstract

Early-exit neural networks (EENNs) enable adaptive and efficient inference by providing predictions at multiple stages during the forward pass. In safety-critical applications, these predictions are meaningful only when accompanied by reliable uncertainty estimates. A popular method for quantifying the uncertainty of predictive models is the use of prediction sets. However, we demonstrate that standard techniques such as conformal prediction and Bayesian credible sets are not suitable for EENNs. They tend to generate non-nested sets across exits, meaning that labels deemed improbable at one exit may reappear in the prediction set of a subsequent exit. To address this issue, we investigate anytime-valid confidence sequences (AVCSs), an extension of traditional confidence intervals tailored for data-streaming scenarios. These sequences are inherently nested and thus well-suited for an EENN's sequential predictions. We explore the theoretical and practical challenges of using AVCSs in EENNs and show that they indeed yield nested sets across exits. Thus our work presents a promising approach towards fast, yet still safe, predictive modeling.

## 1 INTRODUCTION

Modern predictive models are increasingly deployed to environments in which computational resources are either constrained or dynamic. In the constrained setting, the available resources are fixed and often modest. For example, when models are deployed on low-resource devices such as mobile phones, they need to make fast yet accurate predictions for the sake of the user experience. On the other hand, in the dynamic setting, the available resources can vary due to

external conditions. Consider an autonomous vehicle: when it is moving at high speeds, the model must make rapid predictions. However, as the vehicle slows down, the model can afford more time to process information or 'think'. Early-exit neural networks (EENNs) [Teerapittayanon et al., 2016, Huang et al., 2018] present a promising solution to challenges arising in both of these settings. As the name implies, these architectures have multiple exits that allow a prediction to be generated at an arbitrary stopping time. This is in contrast to traditional NNs that yield a single prediction after processing all layers or model components.

To employ EENNs in safety-critical applications such as autonomous driving, it is necessary to estimate the predictive uncertainty at each exit [McAllister et al., 2017]. One prominent approach to capture a model's predictive uncertainty is constructing prediction sets or intervals.[1] Prediction sets aim to cover the ground-truth label with high probability, and their size measures the model's certainty in its prediction. Prediction sets based on Bayesian methods [Meronen et al., 2024] and conformal prediction [Schuster et al., 2021] have been explored for EENNs. However, no work that has accounted for the fact that prediction sets computed at neighboring exits are *dependent*. A prediction interval at a given exit should be *nested* within the intervals at the preceding exits (see Figure 1). In other words, if a candidate prediction $y_0$ is in the interval at exit $t - 1$ and drops out of the interval at exit $t$, $y_0$ should not re-enter the interval at exit $t + 1$. An even worse case would if the intervals at exit $t$ and $t + 1$ are disjoint. Such non-nested behaviour limits the decisions that can be made at the initial exits of an EENN, thereby undermining their anytime properties [Zilberstein, 1996].

We address this open problem by applying *anytime-valid confidence sequences* (AVCSs) [Robbins, 1967, 1970, Lai, 1976] to the task of constructing prediction sets across the exits of an EENN. AVCSs extend traditional, point-wise confidence intervals to streaming data scenarios [Maharaj

[*]Corresponding author: m.jazbec@uva.nl

---

[1]We use the terms prediction *sets* and prediction *intervals* interchangeably, unless otherwise specified.

*Accepted for the 40th Conference on Uncertainty in Artificial Intelligence* (UAI 2024).

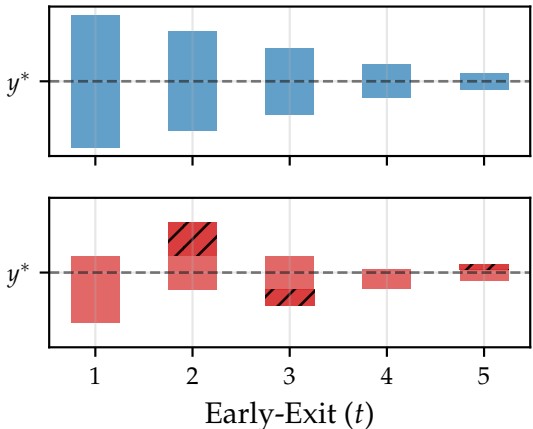

Figure 1: Illustrative example of a 1-dimensional regression problem using an Early-Exit neural network (EENN) with $T = 5$ exits. *Upper*: At each exit, the EENN produces a prediction interval $C_t$ nested within its previous estimates, i.e., $C_t \subseteq C_{t-1}$. *Lower*: An example of non-nested prediction intervals across different exits, e.g., $C_2$ contains candidate labels $y$ not included in $C_1$ (area denoted with (╱) lines). Such behavior often results from an EENN becoming over-confident, i.e., exhibiting low uncertainty, too early.

et al., 2023]. Importantly, AVCSs are guaranteed to have a non-increasing interval width [Howard et al., 2021] and are therefore nested by definition. Our main insight is that AVCSs can be applied (with assumptions) when only one data point is observed, as is the case when constructing the prediction set for a single test point. To achieve this for EENNs, we consider the model parameters (e.g., the output weights) to be 'streaming' across exits. We detail the approximations necessary to make AVCSs applicable for the sequential prediction setting of EENNs and provide bounds on the errors introduced by our approximations. In our experiments across various classification and regression tasks, we demonstrate that our AVCS-based procedure yields nested estimates of predictive uncertainty across the exits of EENNs.

## 2 BACKGROUND

**Data** Let $\mathcal{X} \subseteq \mathbb{R}^D$ denote a $D$-dimensional feature space and $\mathcal{Y}$ the response (output) space. In the case of regression, we have $\mathcal{Y} \subseteq \mathbb{R}$, and for classification $\mathcal{Y} = \{1, \ldots, K\}$. We assume $\boldsymbol{x}$ and $y$ are realizations of the random variables $\mathbf{x}$ and $\mathbf{y}$, drawn from the unknown data distribution $\mathbb{P}(\mathbf{x}, \mathbf{y}) = \mathbb{P}(\mathbf{y}|\mathbf{x}) \, \mathbb{P}(\mathbf{x})$. The training data consists of $N$ feature-response pairs $\mathcal{D} = \{(\boldsymbol{x}_n, y_n)\}_{n=1}^N$. Lastly, let $(\boldsymbol{x}^*, y^*)$ denote a test point, which may be drawn from a different distribution than the one used for training.

**Early-Exit Neural Networks** EENNs [Teerapittayanon et al., 2016, Huang et al., 2018] generate predictions at various depths by having several prediction heads branch out from a shared backbone network. Specifically, an EENN defines a sequence of predictive models: $f(\boldsymbol{x}; \boldsymbol{W}_t, \boldsymbol{U}_{1:t})$, $t = 1, \ldots, T$, where $\boldsymbol{W}_t$ represents the parameters of the predictive head at exit $t$ and $\boldsymbol{U}_t$ denotes the parameters of the $t$-th block in the backbone architecture. EENNs are usually trained by fitting all exits at once $\mathcal{L}(\boldsymbol{W}_{1:T}, \boldsymbol{U}_{1:T}; \mathcal{D}) :=$ $\sum_{n=1}^N \frac{1}{T} \sum_{t=1}^T \ell\big(y_n, f(\boldsymbol{x}_n; \boldsymbol{W}_t, \boldsymbol{U}_{1:t})\big)$ where $\ell$ is a suitable loss function such as negative log-likelihood.

At test time, we can utilize the intermediate predictions of EENNs in various ways. For instance, if the model is deemed sufficiently confident at exit $t$, we can halt computation without propagating through blocks $t + 1, \ldots, T$, thus speeding up prediction time. Naturally, the merit of such an approach relies on quality estimates of the EENN's uncertainty at every exit. EENNs can also be employed as anytime predictors [Zilberstein, 1996, Jazbec et al., 2023]: the aim is to quickly provide an approximate prediction—ideally with its associated uncertainty—and continuously improve upon it as long as the environment permits.

**Prediction Sets** Quantifying the uncertainty of a predictive model $f_\theta : \mathcal{X} \to \mathcal{Y}$ is crucial for its robustness and reliability. A popular approach, which is the focus of this study, augments the model output in the form of a prediction set (or interval, in the case of regression) $C_\theta : \mathcal{X} \to 2^{\mathcal{Y}}$. For a given test point, $C_\theta(\boldsymbol{x}^*)$ should include (or *cover*) the ground-truth $y^*$ with high probability. The *size* of $C_\theta(\boldsymbol{x}^*)$ can be interpreted as a proxy for the model's confidence—a smaller set indicates certainty, a larger set indicates uncertainty. Conformal prediction [Vovk et al., 2005, Shafer and Vovk, 2008] is a popular method to construct prediction sets. Requiring only a calibration dataset $\mathcal{D}_{cal}$, it can generate prediction sets for a given model *post hoc* and with finite-sample, distribution-free guarantees on the coverage of the ground-truth label. See Angelopoulos et al. [2023] for an introduction to conformal prediction. Alternatively, one can employ Bayesian modeling [Gelman et al., 1995] to first obtain a posterior predictive distribution $p(\mathbf{y}|\boldsymbol{x}^*, \mathcal{D})$ and then construct a credible set/interval based on it.

**Anytime-Valid Confidence Sequences** Consider a streaming setting in which new data arrives at every time point $t$ via sampling from an unknown (parametric) model $\boldsymbol{x}_t \sim p(\mathbf{x}|\theta^*)$. Here $\theta^* \in \mathbb{R}$ represents the parameter of the data-generating distribution for which we want to perform statistical inference. An anytime-valid confidence sequence (AVCS) [Robbins, 1967, 1970, Lai, 1976] for $\theta^*$ is a sequence of confidence intervals $C_t = (l_t, r_t) \subseteq \mathbb{R}$ that have time-uniform and non-asymptotic coverage guarantees: $\mathbb{P}(\forall t, \ \theta^* \in C_t) \geq 1 - \alpha$, where $\alpha \in (0, 1)$ represents the level of significance. The anytime (i.e. time-uniform) property allows the user to stop the experiment, 'peek' at

the current results, and choose to continue or not, all while preserving the validity of the statistical inference. This is in contrast with standard confidence intervals based on the central limit theorem (CLT), which are valid only pointwise (i.e. for a fixed time / sample size). The stronger theoretical properties of AVCSs come at a cost, as they are typically larger than CLT-based intervals [Howard et al., 2021].

An AVCS is constructed by first specifying a family of stochastic processes $\{R_t(\theta) : \theta \in \Theta\}$ that depends only on observations $\boldsymbol{x}_1, \ldots, \boldsymbol{x}_t$ available at time $t$. Next, we require that when evaluated at the parameter of interest, $R_t(\theta^*)$ forms a discrete, non-negative *martingale* [Ramdas et al., 2020]—a stochastic process that remains constant in expectation:[2] $\mathbb{E}_{\mathbf{x}_{t+1}}[R_{t+1}(\theta^*)|\mathbf{x}_1, \ldots, \mathbf{x}_t] = R_t(\theta^*), \forall t$. Additionally, $R_0(\theta^*)$ should have an initial value that is constant (usually one). Once such a martingale is constructed, the AVCS at a given $t$ is implemented by computing $R_t(\theta)$ for all $\theta \in \Theta$ and adding to the set the values for which $R_t$ does not exceed $1/\alpha$: $C_t := \{\theta : R_t(\theta) \leq 1/\alpha\}$. Strong theoretical properties (i.e., time-uniformity) then follow from Ville's inequality for nonnegative (super)martingales: $\mathbb{P}(\exists t : R_t(\theta^*) \geq 1/\alpha) \leq \alpha$. One example of a random variable $R_t$ from which we can construct an AVCS is the prior-posterior ratio: $R_t(\theta) = p(\theta)/p(\theta|\boldsymbol{x}_1, \ldots, \boldsymbol{x}_t)$ [Waudby-Smith and Ramdas, 2020]. The time-uniform nature of AVCSs enables one to consider the intersection of all previous intervals—$C_t = \cap_{s \leq t} C_s$, at time $t$—without sacrificing statistical validity [Shekhar and Ramdas, 2023]. This results in nested intervals/sets, i.e., $C_t \subseteq C_{t-1}$. We wish to exploit this pivotal property of AVCSs to ensure that the prediction sets of EENNs remain nested across exits.

# 3 CONFIDENCE SEQUENCES FOR EARLY-EXIT NEURAL NETWORKS

Our contribution is to apply AVCSs to perform inference over the predictions generated by each exit of a EENN. As we will see, this is not a straightforward synthesis: AVCSs have been exclusively used in streaming-data settings, where the goal at every time step is to produce a *confidence interval* covering the parameter of the data generating distribution $\theta$. On the other hand, we want to apply them to EENNs that see just one feature vector $\boldsymbol{x}^*$ at test time. Moreover, we are interested in obtaining a *prediction set/interval* at every exit that contains the ground-truth label $y^*$ with high probability. We overcome these differences by considering the parameters of the EENN's exits $\boldsymbol{W}_t$ as the sequence of random variables for which the martingale is defined. Below we first give a general recipe for constructing AVCSs for EENNs and then describe practical implementations for regression (Section 4) and classification (Section 5).

**Bayesian EENN** We begin by positing a (last-layer) Bayesian predictive model at every exit:[3]

$$p_t(\mathrm{y}|\boldsymbol{x}^*, \mathcal{D}) = \int p(\mathrm{y}|\boldsymbol{x}^*, \mathbf{W}_t, \boldsymbol{U}_{1:t}) \, p(\mathbf{W}_t|\mathcal{D}, \boldsymbol{U}_{1:t}) \, d\mathbf{W}_t \tag{1}$$

for $t = 1, \ldots, T$, with $T$ representing the total number of exits. $p(\mathrm{y}|\boldsymbol{x}^*, \mathbf{W}_t, \boldsymbol{U}_{1:t})$ and $p(\mathbf{W}_t|\mathcal{D}, \boldsymbol{U}_{1:t})$ correspond to the likelihood and (exact) posterior distribution, respectively. To ensure minimal overhead of our approach at test time, we treat the backbone parameters $\boldsymbol{U}_{1:t}$ as point estimates (e.g. found through pre-training) that are held constant when constructing the AVCS. To reduce notational clutter, we omit these parameters from here forward. While Bayesian predictives $p_t(\mathrm{y}|\boldsymbol{x}^*, \mathcal{D})$ can be used 'as is' to get uncertainty estimates at each exit (e.g., by constructing a credible interval), we show in Section 7 that this results in a non-nested sequence of uncertainty estimates. We next present an approach based on AVCSs to rectify such behaviour.

**Idealized Construction** We first consider an idealized construction that, while impossible to implement exactly, will serve as the foundation of our approach. At test time, upon seeing a new feature vector $\boldsymbol{x}^*$, we wish to compute an interval $C_t$ for its label such that $y^* \in C_t \,\forall t$ with high probability. Assume that we also have observed the true label $y^*$. For the moment, ignore the circular reasoning that this is the very quantity for which we wish to perform inference. Furthermore, with $(\boldsymbol{x}^*, y^*)$ in hand, assume we can compute (exactly) the posterior for any exit's parameters: $p(\mathbf{W}_t|, \mathcal{D} \cup (\boldsymbol{x}^*, y^*))$. This distribution is the posterior update we would perform after observing the new feature-response pair. For notational brevity, we will denote $\mathcal{D}_* := \mathcal{D} \cup (\boldsymbol{x}^*, y^*)$ from here forward.

To prepare for the proposition that follows, we define for a given $y \in \mathcal{Y}$ the *predictive-likelihood ratio*

$$R_t^*(y) := \prod_{l=1}^{t} \frac{p_l(y|\boldsymbol{x}^*, \mathcal{D})}{p(y|\boldsymbol{x}^*, \boldsymbol{W}_l)}, \quad \boldsymbol{W}_l \sim p(\mathbf{W}_l|\mathcal{D}_*). \tag{2}$$

Note that only the likelihood terms in the denominator depend on the updated posterior (via samples $\boldsymbol{W}_l$), whereas the predictive terms in the numerator rely solely on training data (via $p(\mathbf{W}_l|\mathcal{D})$). The above ratio in (2) is inspired by the aforementioned prior-posterior martingale [Waudby-Smith and Ramdas, 2020] yet modified for the predictive setting. We next state our key proposition that will serve as an inspiration for constructing AVCS for $y^*$ in EENNs:

---

[2]It is also common to define AVCS in terms of *supermartingales*, which are stochastic processes that decrease in expectation over time: $\mathbb{E}_{\mathbf{x}_{t+1}}[R_{t+1}(\theta^*)|\mathbf{x}_1, \ldots, \mathbf{x}_t] \leq R_t(\theta^*), \forall t$.

[3]In this section, we work with Bayesian predictive models at every exit for ease of exposition. Yet our approach is more general. It can also accommodate models for which the 'randomness' does not come from placing a distribution over weights $\mathbf{W}_t$. We will provide a concrete example of this later in Section 5, where we use an evidential approach [Malinin and Gales, 2018, Sensoy et al., 2018] instead of a Bayesian one.

**Proposition 1.** *For a given test point $(\boldsymbol{x}^*, y^*)$, the predictive-likelihood ratio $R_t^*(y)$ in (2) is a non-negative martingale with $R_0^* = 1$ when evaluated at $y = y^*$. Moreover, the prediction sets of the form $C_t^* := \{y \in \mathcal{Y} \,|\, R_t^*(y) \leq 1/\alpha\}$ are $(1-\alpha)$-confidence sequences for $y^*$, meaning that $\mathbb{P}(\forall t, y^* \in C_t^*) \geq 1 - \alpha$.*

The proof follows the standard procedure for deriving parametric confidence sequences; see Appendix B.1. We term the resulting confidence sequence an *EENN-AVCS*.

**Realizable Relaxation** Now we return to the aforementioned circular reasoning: we are performing inference for $y^*$ while assuming we have access to it. In practice, we do not have access to $y^*$ at test time; hence we cannot compute $R_t^*(y)$ (and consequently $C_t^*$). As a workaround, we propose to approximate the updated posterior with the one based on only the training data at every exit $t = 1, \ldots, T$:

$$p(\mathbf{W}_t | \mathcal{D}_*) \approx p(\mathbf{W}_t | \mathcal{D}). \tag{3}$$

With $R_t(y)$ and $C_t$, we denote the resulting predictive-likelihood ratio and confidence sequence based on $p(\mathbf{W}_t | \mathcal{D})$, respectively. While $C_t$ is now computable in a real-world scenario (since it is independent of $y^*$), it unfortunately does not inherit the statistical validity of $C_t^*$. Naturally, the degree to which $C_t$ violates validity depends on the quality of approximation in (3). If the posterior distribution $p(\mathbf{W}_t | \mathcal{D})$ is stable—meaning that adding a single new data point $(\boldsymbol{x}^*, y^*)$ would have minimal effect—the approximation is well-justified, and only minor validity violations can be expected. Such stability in the posterior is likely when the training dataset $\mathcal{D}$ is large and the new test datapoint originates from the same distribution. Conversely, if the posterior is unstable, the approximation will likely be poor, leading to larger violations of validity. This intuition can be formalized via the following proposition:

**Proposition 2.** *Assume $C_t^*$ is a valid $(1 - \alpha)$ confidence sequence for a given test datapoint $(\boldsymbol{x}^*, y^*)$ (c.f. Proposition 1). Then the miscoverage probability of the confidence sequence $C_t := \{y \in \mathcal{Y} \mid R_t(y) \leq 1/\alpha\}$ can be upper bounded by*

$$P(\exists l \in \{1, \ldots, t\}, y^* \notin C_l) \leq$$
$$\alpha + \sqrt{1 - e^{-\sum_{l=1}^{t} KL\left(p(\mathbf{W}_l | \mathcal{D}), \, p(\mathbf{W}_l | \mathcal{D}_*)\right)}}$$

*$\forall t = 1, \ldots, T$, where $KL$ denotes the Kullback-Leibler divergence between probability distributions.*

See Appendix B.2 for the derivation. Based on the bound in Proposition 2, it is clear that when the posteriors at different exits are stable, i.e. the KL divergence between $p(\mathbf{W}_l | \mathcal{D})$ and $p(\mathbf{W}_l | \mathcal{D}_*)$ is small, the validity violation is minor. As a result, $C_t$ will be a good approximation of $C_t^*$.

**Detecting Violations of Posterior Stability** It is evident from Proposition 2 that when the approximation in (3) is poor—i.e. the KL divergence between $p(\mathbf{W}_l | \mathcal{D})$ and $p(\mathbf{W}_l | \mathcal{D}_*)$ is large—the validity of $C_t$ will quickly degrade. As aforementioned, this could happen for a particular $\boldsymbol{x}^*$ if either (i) $\mathcal{D}$ is small and the posterior is not stable yet or (ii) $\boldsymbol{x}^*$ is not drawn from the training distribution. The method should fail gracefully in such cases. Fortunately, the behavior of invalid AVCSs—ones for which $R_t(y)$ is not a martingale for all $y \in \mathcal{Y}$—has been previously studied for change-point detection [Shekhar and Ramdas, 2023]. Based off of their theoretical and empirical results, our procedure should collapse to the empty interval if the approximation (3) is poor: $\exists t_0$ such that $C_{t \geq t_0} = \emptyset$. Encouragingly, in Section 7.1, we experimentally validate that such collapses occur for out-of-distribution points for a reasonably small $t_0$. However, there will be times at which the interval width will be small—which the user might interpret as high confidence—only to later collapse to the empty set (meaning maximum uncertainty). In Section 7.1, we explore using epistemic uncertainty as a measure of stability in our regression models, and we leave to future work a more general method for diagnosing when an EENN-AVCS has not yet collapsed but is likely to.

# 4 EENN-AVCS FOR REGRESSION

We next consider a concrete instantiation of our EENN-AVCS procedure proposed in the previous section. We focus on the case of one-dimensional Bayesian regression as it allows for exact inference due to conjugacy. This allows us to assess the quality of approximation (3) without introducing the additional challenge of approximate inference. We summarize our approach for obtaining AVCSs in EENNs in Algorithm 1.

**Bayesian Linear Regression** Recall from Section 3 that since we require fast and exact Bayesian inference, we keep EENN's backbone parameters $\boldsymbol{U}_t$ fixed and give only the weights $\boldsymbol{W}_t$ of the prediction heads a Bayesian treatment. We define the predictive model at the $t$th exit as a linear model $f(\boldsymbol{x}; \boldsymbol{W}_t, \boldsymbol{U}_{1:t}) = h_t(\boldsymbol{x})^T \boldsymbol{W}_t$ where $h_t(\,\cdot\,; \boldsymbol{U}_{1:t}) : \mathcal{X} \to \mathbb{R}^H$ represents the output of the first $t$ backbone layers or blocks. We use a Gaussian likelihood and prior:

$$\mathrm{y} \sim \mathcal{N}\left(\mathrm{y}; h_t(\boldsymbol{x})^T \boldsymbol{W}_t, \sigma_t^2\right), \quad \boldsymbol{W}_t \sim \mathcal{N}\left(\boldsymbol{W}_t; \hat{\boldsymbol{W}}_t, \sigma_{w,t}^2 \mathbb{I}_H\right)$$

where $\sigma_t^2$ is the observation noise, $\sigma_{w,t}^2$ is the prior's variance, and $\hat{\boldsymbol{W}}_t$ are the prediction weights obtained during (pre)training of the EENN. Due to conjugacy, we can obtain a closed form for the posterior and predictive distributions:

$$p(\mathbf{W}_t | \mathcal{D}) = \mathcal{N}\left(\boldsymbol{W}_t; \bar{\boldsymbol{\mu}}_t, \bar{\boldsymbol{\Sigma}}_t\right),$$
$$p_t(\mathrm{y} | \boldsymbol{x}^*, \mathcal{D}) = \mathcal{N}\left(\mathrm{y}; h_t(\boldsymbol{x}^*)^T \bar{\boldsymbol{\mu}}_t, v_* + \sigma_t^2\right), \tag{4}$$

where $v^* := h_t(\boldsymbol{x}^*)^T \bar{\boldsymbol{\Sigma}}_t h_t(\boldsymbol{x}^*)$. See Appendix B.3 for exact expressions for posterior parameters $\bar{\boldsymbol{\mu}}_t, \bar{\boldsymbol{\Sigma}}_t$. To estimate $\sigma_t^2$ and $\sigma_{w,t}^2$, we optimize the (exact) marginal likelihood on the training data (type-II maximum likelihood). Combining the obtained Bayesian quantities, we can compute the predictive-likelihood ratio in (2) at every exit.

**Solving for Interval Endpoints** To construct $C_t$, we next have to evaluate $R_t$ at every $y \in \mathcal{Y}$ and discard those where the ratio exceeds $1/\alpha$, with $\alpha$ representing a significance level (e.g., 0.05). However, in the case of regression, where the output space is continuous, the method of evaluation is not immediately clear. One possible approach would be to define a grid of points over $\mathcal{Y}$ and then evaluate the predictive-likelihood ratio using a finite number of labels. Fortunately, the Bayesian linear regression model above allows us to obtain the endpoints of the prediction interval, at all exits, via a closed-form expression: $C_t = [y_L^t, y_R^t]$. This is computationally valuable since it eliminates the overhead of iterating over $\mathcal{Y}$, which could be prohibitively expensive in the low-resource settings in which EENNs typically operate. To arrive at the analytical form, we first observe that $\log R_t$ represents a convex quadratic function in $y$:

$$\log R_t(y) = \alpha_t(\boldsymbol{x}^*) \cdot y^2 + \beta_t(\boldsymbol{x}^*, \boldsymbol{W}_{1:t}) \cdot y + \gamma_t(\boldsymbol{x}^*, \boldsymbol{W}_{1:t}).$$

Expressions for the coefficients $\alpha_t, \beta_t, \gamma_t$ are provided in Appendix B.4. To obtain the bounds $y_L^t, y_R^t$ of the prediction interval at the $t$th exit, we then simply need to find the roots of the quadratic equation $\log R_t(y) - \log(1/\alpha) = 0$. If the discriminant $\beta_t^2 - 4\alpha_t(\gamma_t + \log \alpha)$ is negative, the equation has no real-valued roots, resulting in an empty prediction interval. In such cases, we interpret $\boldsymbol{x}^*$ as an out-of-distribution sample, as mentioned in Section 3.

**Epistemic Uncertainty as a Measure of Stability** In our assumed Bayesian linear regression scenario, both the posterior and updated posterior are Gaussian. This allows us to derive a closed-form expression for the KLD term $KL\big(p(\mathbf{W}_t|\mathcal{D}), p(\mathbf{W}_t|\mathcal{D}_*)\big)$ in the upper bound from Proposition 2. See Appendix B.5 for the derivation. Recall that $v_*$ represents the epistemic uncertainty (c.f. Eq. (4)), which is the uncertainty that stems from observing limited data. In turn, the KLD is small for a given $\boldsymbol{x}^*$ when $v_*$ is small. The uncertainty decreases as we collect more data[4], which, together with Proposition 2, implies that the statistical coverage of our EENN-AVCS will improve as the dataset size increases. Moreover, $v^*$ is independent of the test label $y^*$. Thus, we can employ it as a measure of the stability of a EENN-AVCS: for a given $\boldsymbol{x}_*$, a higher $v^*$ can signal to the user that the resulting confidence sequence may not be reliable. We illustrate this in Section 7.1.

# 5 EENN-AVCS FOR CLASSIFICATION

In this section, we propose a concrete instantiation of our EENN-AVCS for classification. Unlike the regression scenario in the previous section, an additional challenge is presented by a lack of conjugacy. Specifically, we cannot obtain a closed-form expression for the Bayesian predictive posterior (see Eq. (1)) at every exit when using the usual Gaussian assumption for the posterior over parameters. To circumvent this, we depart from the Bayesian predictive model and utilize instead Dirichlet Prior Networks [Malinin and Gales, 2018], which enable analytically tractable predictive distributions at each exit. Our EENN-AVCS approach for classification is summarized in Algorithm 2.

**Dirichlet Prior Networks** Instead of positing a distribution over (last-layer) weights $\mathbf{W}_t$ at every exit, we posit a distribution over categorical distributions $p(\boldsymbol{\pi}_t|\mathcal{D}, \boldsymbol{x}^*)$, $\boldsymbol{\pi}_t \in \Delta^K$ [5] for a given test datapoint $\boldsymbol{x}^*$. Assuming a categorical likelihood, the posterior is Dirichlet via conjugacy:

$$p(\mathbf{y}|\boldsymbol{\pi}_t) = \texttt{Cat}(\mathbf{y}|\boldsymbol{\pi}_t), \ p(\boldsymbol{\pi}_t|\boldsymbol{x}^*, \mathcal{D}) = \texttt{Dir}(\boldsymbol{\pi}_t|\boldsymbol{\alpha}_t(\boldsymbol{x}^*; \mathcal{D}))$$

where $\boldsymbol{\alpha}_t \in \mathbb{R}_{>0}^K$ are the concentration parameters. The predictive distribution also has a closed form:

$$p_t(\mathbf{y} = y|\boldsymbol{x}^*, \mathcal{D}) =$$
$$\int p(\mathbf{y} = y|\boldsymbol{\pi}_t) \, p(\boldsymbol{\pi}_t|\boldsymbol{x}^*, \mathcal{D}) \, d\boldsymbol{\pi}_t = \frac{\alpha_{t,y}}{\sum_{y' \in \mathcal{Y}} \alpha_{t,y'}} \ .$$

Malinin and Gales [2018] propose to parameterize the Dirichlet concentration parameters via the outputs of a neural network, $\boldsymbol{\alpha}_t(\boldsymbol{x}^*; \mathcal{D}) = f(\boldsymbol{x}^*; \boldsymbol{W}_t, \boldsymbol{U}_{1:t})$, and term this model a *Dirichlet Prior Network* (DPN). In DPNs, the aim is to capture the *distributional uncertainty* that arises due to the mismatch between test and training distributions, in addition to the *data uncertainty* (often referred to as aleatoric uncertainty). This is in contrast to Bayesian models, which focus on the *model uncertainty* (or epistemic uncertainty). We refer the reader to Malinin and Gales [2018] for an in-depth discussion of the different sources of uncertainty.

**Classification EENN-AVCS** Having a closed-form predictive distribution, we can define the following *predictive-likelihood* ratio for a given $y \in \mathcal{Y}$:

$$R_t^*(y) := \prod_{l=1}^t \frac{p_l(y|\boldsymbol{x}^*, \mathcal{D})}{p(y|\boldsymbol{\pi}_l)}, \ \boldsymbol{\pi}_l \sim p(\boldsymbol{\pi}_l|\mathcal{D}^*) \ .$$

Our result from Proposition 1 applies here as well[6], hence it follows that $C_t^* := \{y \in \mathcal{Y} \mid R_t^*(y) \le 1/\alpha\}$ is a valid

---

[4]$\lim_{N \to \infty} v_* = 0$ where $N$ represents the number of training data points (c.f. Section 3.3.2 in Bishop and Nasrabadi [2006]).

[5]$\Delta^K := \{\boldsymbol{\pi} \in \mathbb{R}^K \mid \sum_{k=1}^K \pi_k = 1, \pi_k \ge 0\}$
[6]The only difference in the proof being that the martingale is defined with respect to the sequence of categorical distributions $\boldsymbol{\pi}_t$ instead of the sequence of weights $\mathbf{W}_t$.

$(1 - \alpha)$-confidence sequences for $y^*$. As in the regression case, $R_t^*$ can not be realized in practice as it depends on the unknown label $y^*$. We again approximate this oracle posterior with the one based solely on the training data $p(\boldsymbol{\pi}_l | \mathcal{D}^*) \approx p(\boldsymbol{\pi}_l | \boldsymbol{x}^*, \mathcal{D})$ and denote the resulting predictive-likelihood ratio and confidence sequence as $R_t$ and $C_t$, respectively. To reason about the quality of this approximation, we can again rely on Proposition 2.

**Post-Hoc Implementation**   The original DPN formulation [Malinin and Gales, 2018] requires a specialized training procedure to ensure that the NN's outputs represent meaningful concentration parameters. We instead opt for a simpler *post-hoc* approach as we have found it to yield satisfactory results. Specifically, to obtain the concentration parameters, we start with a pretrained (classification) EENN and pass the logits at each exit through an activation function $a : \mathbb{R} \to \mathbb{R}_{>0}$. We found that a simple choice of ReLU activation $a_t(x) = \text{ReLU}(x, \tau_t)$ with a different threshold $\tau_t \geq 1$ at each exit works well in practice.[7] To obtain the ReLU thresholds, we use a validation dataset and pick the largest $\tau_t$ such that $(1-\alpha)\%$ of validation datapoints are still contained in the resulting prediction sets at each exit. Lastly, since $\mathcal{Y}$ has a finite support (unlike the regression case), we iterate over all of $\mathcal{Y}$ when constructing a prediction set $C_t$.

# 6   RELATED WORK

**Early-Exit Neural Networks** (EENNs) enable faster inference in deep models by allowing predictions to be made at intermediate layers [Teerapittayanon et al., 2016, Huang et al., 2018, Laskaridis et al., 2021]. They have been extensively explored for computer vision [Li et al., 2019, Kaya et al., 2019, Yang et al., 2023] and natural language processing [Schwartz et al., 2020, Zhou et al., 2020, Xu and McAuley, 2023]. The majority of these studies aimed to improve the accuracy-speed trade-off, i.e., ensuring the model exits as early as possible while maintaining high accuracy. However, uncertainty quantification (UQ) within EENNs has so far received relatively little attention [Schuster et al., 2021, Meronen et al., 2024, Regol et al., 2024]. When it has, UQ has primarily been used to improve EENN termination criteria. Meronen et al. [2024] employ a Bayesian predictive model at each exit to enhance the calibration of EENNs. Schuster et al. [2021] propose a conformal prediction scheme with the goal of generating sets/intervals that are (marginally) guaranteed to contain the prediction of the full EENN. Yet none of the preceding works address the fact that uncertainty estimates at successive exits are dependent, which is the main focus of our work. Perhaps the closest related work is by Jazbec et al. [2023], who adapt EENNs for

the anytime setting [Zilberstein, 1996]. Their method promotes *conditional monotonicity*: the EENN's performance improves across exits for every test sample. Our idea of nested prediction sets can be seen as an extension of conditional monotonicity to EENNs that yield prediction sets, not only point predictions as done by Jazbec et al. [2023].

**Anytime-Valid Confidence Sequences** (AVCSs) are sequences of confidence intervals designed for streaming data settings, providing time-uniform and non-asymptotic coverage guarantees [Robbins, 1967, Lai, 1976, Howard et al., 2021]. They allow for adaptive experimentation that permits one to 'peek' at the data at any time, make decisions, yet still maintain the validity of the statistical inferences. Recently, AVCSs have found applications in A/B testing that is resistant to 'p-hacking' [Maharaj et al., 2023], Bayesian optimization [Neiswanger and Ramdas, 2021], and change-point detection [Shekhar and Ramdas, 2023]. AVCSs have not been previously considered for sequential estimation of predictive uncertainty in EENNs.

# 7   EXPERIMENTS

We conduct three sets of experiments, which can be reproduced using the code at `https://github.com/metodj/EENN-AVCS`. Firstly, in Section 7.1, we explore our method (EENN-AVCS) on synthetic datasets to empirically verify its correctness and assess its feasibility. In the subsequent set of experiments, detailed in Section 7.2, we check that our findings extend to practical scenarios, applying EENN-AVCS to a textual semantic similarity regression task using a transformer backbone model [Zhou et al., 2020]. Lastly, in Section 7.3, we report results on image classification tasks (CIFAR-10/100, ImageNet) using a multi-scale dense net (MSDNet) [Huang et al., 2018].

**Evaluation Metrics**   To assess the quality of the prediction sets at each exit, we utilize the standard combination of *marginal coverage* and *efficiency*, i.e. average interval size, on the test dataset [Angelopoulos et al., 2023]:

$$\text{size}(t) := \frac{1}{n_{test}} \sum_{n=1}^{n_{test}} |C_t(\boldsymbol{x}_n)|,$$

$$\text{coverage}(t) := \frac{1}{n_{test}} \sum_{n=1}^{n_{test}} \big[ y_n \in C_t(\boldsymbol{x}_n) \big],$$

where $C_t$ is a prediction set at the $t$-th exit and $[\cdot]$ is the indicator function. Marginal coverage serves as a proxy for the statistical validity of the approach, measuring how frequently the ground-truth falls within the predicted interval on average. Among two methods with similar marginal coverage, the one with smaller interval sizes is preferred. To assess the nestedness of prediction sets across exits, we define a *nestedness* metric: at each exit $t$, we compute

$$\mathfrak{N}(t) = |\cap_{s \leq t} C_s| / |C_t|$$

---

[7]We restrict concentration parameters to be larger than one due to the Dirichlet concentrating towards the simplex's edges for parameter values smaller than one.

and report its mean across test data points. A model with perfectly nested prediction sets will have $\mathfrak{N}(t) = 1$, exactly. Otherwise, $\mathfrak{N}(t)$ will be less than one and zero only in the case of disjoint sets.

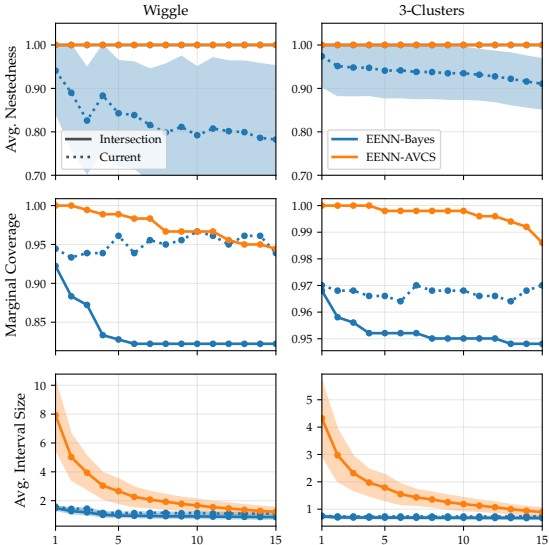

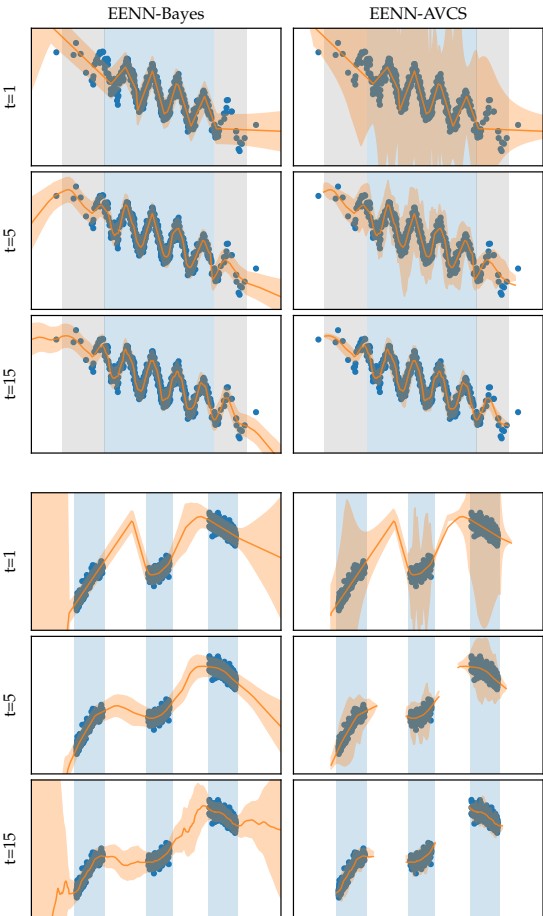

Figure 2: We compare our EENN-AVCS with EENN-Bayes baseline based on average nestedness (*top*), marginal coverage (*middle*), and average interval size (*bottom*). EENN-AVCS is the only approach that yields perfect nestedness while maintaining reasonably high marginal coverage across exits. The nestedness comes at a price of larger intervals in the initial exits, though. Note that in the *top* plot, the nestedness curves of EENN-AVCS (—) and EENN-Bayes-intersection (—) overlap at $\mathfrak{N}(t) = 1$.

**Baselines** We compare EENN-AVCS against standard UQ techniques—namely Bayesian methods and conformal prediction. As a Bayesian baseline, we use the same underlying Bayesian EENN but without applying the AVCS. We term this approach *EENN-Bayes* since it uses the Bayesian predictive distribution at each exit to perform UQ. EENN-Bayes can be seen as an adaptation of the last-layer Laplace approach for early-exiting [Meronen et al., 2024]. For the conformal baselines, we perform conformal inference independently at every exit. Specifically, we use the Regularized Adaptive Predictive Sets algorithm [RAPS; Angelopoulos et al., 2021] for the classification experiments (c.f., 7.3) and Conformalized Quantile Regression [CQR; Romano et al., 2019] for the NLP regression experiments (c.f., Sec 7.2). The primary difference between our approach and the baselines should be that EENN-AVCS has nested intervals, without sacrificing coverage, whereas the baselines have no such guarantee.

Figure 3: Prediction intervals (■) for EENN-Bayes (*left*) and our EENN-AVCS (*right*) on two simulated regression tasks Antorán et al. [2020]: wiggle (*up*) and 3-clusters (*bottom*). Blue points denote training data. In cases where the EENN-AVCS collapses to an empty set (out-of-distribution), we do not depict anything, which explains the gaps in EENN-AVCS predictions. We set the significance level to $\alpha = 0.05$ for EENN-AVCS, while for EENN-Bayes, we plot intervals that capture 2 standard deviations away from the predicted mean (—). With different background colors we denote different regions of data distribution, see Section 7.1.

## 7.1 SYNTHETIC REGRESSION DATA

We use two non-linear regression simulations [Antorán et al., 2020]: *wiggle* and *3-clusters*. The EENN used in this experiment has a backbone architecture of $T = 15$ feed-forward layers with residual connections. Each layer consists of $M = 20$ hidden units, and we attach an output layer on top of it to enable early-exiting. We fit the (last-layer) Bayesian linear regression model at each exit using the training data and construct $S = 10$ confidence sequences in parallel at test time for each datapoint (see Appendix A.1 for more details on the parallel construction). We set the significance level to $\alpha = 0.05$ for EENN-AVCS, while for EENN-Bayes, we plot intervals that capture two standard deviations away from the predicted mean. Further details regarding data gen-

eration, the model architecture, and the training can be found in Appendix C.1.

In the *top* row of Figure 2, we compare our EENN-AVCS (—) against the EENN-Bayes (⋯) baseline on the test dataset based on how nested the prediction intervals are across exits. We observe that, due to their theoretical foundation, EENN-ACVSs attain perfect nestedness. In contrast, EENN-Bayes's nestedness deteriorates over time on both datasets considered, indicating that there are labels that re-enter the EENN-Bayes prediction intervals after being ruled out at some earlier exit(s). In the *top* row, we additionally observe that perfect nestedness can be achieved in EENN-Bayes by considering a running intersection of all previous prediction intervals at each exit (denoted with (—) line), similar to EENN-AVCS (the two nestedness lines of both intersection methods overlap at $\mathfrak{N}(t) = 1$). However, as shown in the *middle* row, this approach leads to a decrease in marginal coverage, indicating that fewer data points are covered by the intersection of EENN-Bayes intervals as more exits are evaluated. In contrast, EENN-AVCS maintains high marginal coverage despite utilizing an intersection of intervals at each exit. This is a direct consequence of the time-uniform nature of AVCS. The nestedness of EENN-AVCS comes at a price, though, as the interval size tends to be larger than that of EENN-Bayes at the initial exits (*bottom* plot). This observation is in line with existing work on AVCSs [Howard et al., 2021].

To better understand our method's behavior on in-distribution (ID) vs out-of-distribution (OOD) points, we construct a new test dataset by considering equidistantly spaced points across the entire $\mathcal{X}$ space[8]. We report results for both datasets considered in Figure 3. Initially, we observe that for ID datapoints (with ID regions of $\mathcal{X}$ depicted using ■ background), our method satisfactorily covers the data distribution, especially at later exits. Encouragingly, AVCSs are also observed to quickly collapse to empty intervals outside of the data distribution (OOD regions are depicted with a white background). Whenever the AVCS collapses to an empty interval, we omit plotting the EENN-AVCS's predictions, showing the collapse via gaps in Figure 3. Recall that in our setting, an empty interval represents that a distribution shift has been detected (i.e. maximal predictive uncertainty), which is exactly the desired behavior in OOD regions.

On the *wiggle* dataset, we also have the opportunity to study the behavior on the so-called in-between (IB) datapoints that reside between ID and OOD regions. We depict the IB region with a ■ background. We observe that our method encounters challenges in this regime to some extent, as the prediction intervals are, counterintuitively, smaller compared to those in the ID region despite the density of observed

training datapoints being lower in the IB area. A partial remedy is provided by the epistemic uncertainty $v^*$ (see Eq. (4)), which in our framework can be interpreted as a proxy for the stability of posterior distributions at different exits as explained in Section 4. As depicted in Figure 4, $v^*$ is larger for IB points compared to the ID ones (as expected). Thus, a higher $v^*$ can serve as a warning that the resulting confidence sequence should not be blindly relied upon.[9]

[8]Specifically, for $\mathcal{X} = [L, R]$, we construct $X_{test} = $ `np.linspace`$(L - \epsilon, R + \epsilon, N_{test})$ for $\epsilon > 0$.

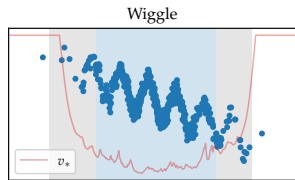
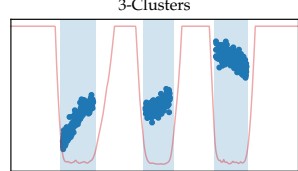

Figure 4: Average epistemic uncertainty $v_*$ (—) across Bayesian linear regression models at different exits. As expected, $v_*$ is larger in the regions where we observe less training data: *out-of-distribution* (denoted with a white background) and *in-between* (denoted with a grey background ■). Hence, $v_*$ can serve as an indicator for assessing the reliability of EENN-AVCSs.

## 7.2 SEMANTIC SIMILARITY USING ALBERT

In this experiment, we examine the STS-B dataset from the GLUE Benchmark [Wang et al., 2019] and the SICK dataset [Marelli et al., 2014]. For both, the task is predicting the degree of semantic similarity between two input sentences. The similarity score is a continuous label ranging between 0 and 5, denoted as $\mathcal{Y} = [0, 5]$. As the backbone model, we employ ALBERT with 24 transformer layers [Lan et al., 2020], providing the model an option to early exit after every layer. Bayesian linear regression models are fitted on the development set. At test time, we construct a single AVCS ($S = 1$) with $\alpha = 0.05$. We observed that constructing multiple AVCSs in parallel leads to a quicker decay of marginal coverage on this dataset. Since we know that the true label is within $[0, 5]$, we clip the resulting prediction intervals for all approaches to this region (if they should extend beyond it). Refer to Appendix C.2 for additional details on data, model, and training for this experiment.

Results are presented in Figure 5. Encouragingly, the observations here align qualitatively with those made on synthetic datasets in Section 7.1. In the *top* plot, considering only the current Bayesian (⋯) or conformal (⋯) interval at each exit again results in non-nested uncertainty estimates. As shown in the *middle* plot, using the running intersection of EENN-Bayes 's (—) and CQR's (—) intervals rectifies this

non-nestedness. However, using the running intersection results in a larger decay in marginal coverage. EENN-AVCS's (—) coverage does not suffer nearly to the same extent. The marginal coverage in the case of the STS-B dataset is worse across all approaches when compared to the coverage observed on synthetic data experiments, c.f. Figure 2. We attribute this to there being a larger shift between training, development, and test data splits for the STS-B dataset, as evidenced by the difference in model performance on each of those splits (see Appendix C.2 for further details). Finally, the *bottom* plot reaffirms that the nestedness of EENN-AVCS comes at the expense of larger intervals.

[9]The IB region also poses challenges for other UQ methods; a similar behavior was reported for Gaussian processes [Lin et al., 2023], with the in-between region being referred to as the *extrapolation region*.

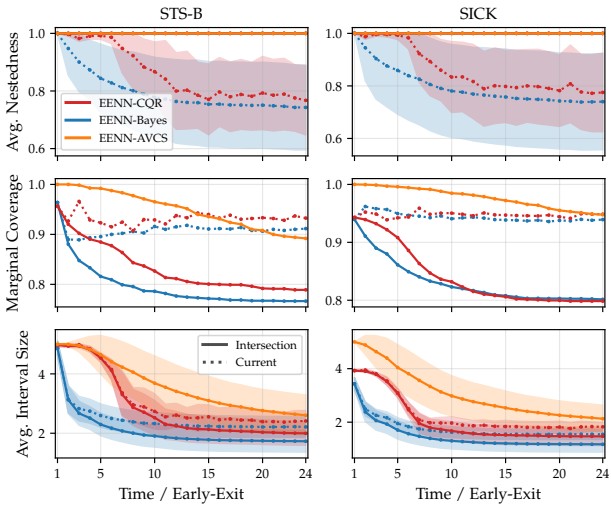

Figure 5: Comparison of our EENN-AVCS with CQR [Romano et al., 2019] and EENN-Bayes baselines on the NLP regression datasets. Similar to findings on the synthetic data (c.f., Figure 2), EENN-AVCS attains perfect nestedness (*upper* plot) while maintaining reasonably high marginal coverage across exits (*middle* plot). However, the intervals generated by EENN-AVCS at each exit are larger compared to the baseline (*bottom* row). Note that in the *upper* plot, the nestedness curves of EENN-AVCS (—), EENN-Bayes-intersection (—), and EENN-CQR-intersection (—) overlap at $\mathfrak{N}(t) = 1$.

## 7.3 IMAGE CLASSIFICATION WITH MSDNET

In the last experiment, we quantify uncertainty at every exit on an image classification task. We consider CIFAR-10/100, [Krizhevsky et al., 2009], and ILSVRC 2012 (ImageNet; Deng et al. [2009]). As our backbone EENN, we employ a Multi-Scale Dense Network [MSDNet; Huang et al., 2018], which consists of stacked convolutional blocks. At each exit, we map the logits to concentration parameters of the Dirichlet distribution using the ReLU activation function, as discussed in Section 5. To find the exact ReLU thresholds at each exit, we allocate 20% of the test dataset as a validation dataset and evaluate the performance on the remaining 80%.

We construct a single AVCS ($S = 1$) at each exit. We use significance level $\alpha = 0.05$ for EENN-AVCS as well as for both baselines.

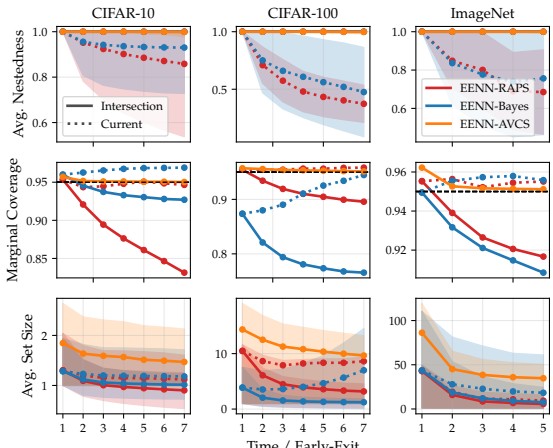

Figure 6: Comparison of our EENN-AVCS with RAPS [Angelopoulos et al., 2021] and EENN-Bayes baselines based on average nestedness (*top*), marginal coverage (*middle*), and average interval size (*bottom*) for our image classification experiments using MSDNet as a backbone. EENN-AVCS is the only approach that attains perfect nestedness (*top*) while maintaining high marginal coverage across different exits (*middle*). Nestedness comes at a price, though, as EENN-AVCS sets are larger compared baseline ones (*bottom*). Note that in the *top* plot, the nestedness curves of EENN-AVCS (—), RAPS-intersection (—), and EENN-Bayes -intersection (—) overlap at $\mathfrak{N}(t) = 1$.

In Figure 6, we observe that constructing conformal RAPS (···) or Bayesian credible (···) sets at every exit independently leads to non-nested behavior (see *top* row). Taking the intersection of RAPS sets (—) corrects this; however, as expected this leads to a violation of conformal marginal coverage guarantees (see *middle* row). The same observations hold for the intersection of EENN-Bayes sets (—). Encouragingly, as in our regression experiments, our EENN-AVCS based on the Dirichlet Prior Network (—) yields perfect nestedness while maintaining high marginal coverage. In the *bottom* row, we also see that EENN-AVCS sets are roughly two times (or less) larger than the sets from both baselines, which might be a reasonable price to pay for the nestedness.

## 8 CONCLUSION

We proposed using anytime-valid confidence sequences for predictive uncertainty quantification in EENNs. We showed that our approach yields nested prediction sets across exits— a property that is lacking in prior work, yet is crucial when deploying EENNs in safety critical applications. We described the theoretical and practical challenges associated with using AVCSs for predictive tasks. Moreover, we empir-

ically validated our approach across a range of EENNs and datasets. Our work is an important step towards models that are not only fast but also safe.

**Limitations and Future Work**  For future work, it is paramount to improve the efficiency of EENN-AVCSs, aiming for smaller intervals. This is especially crucial for the initial exits, which are of the highest practical interest for resource-constrained settings. While we explored ways to reduce the set size (c.f., Appendix A.1), further efforts are necessary to ensure faster convergence without sacrificing marginal coverage in the process. Additionally, studying alternatives to the predictive-likelihood ratio (c.f., Eq. (2)) for constructing confidence sequences might be a promising way to improve efficiency. Finally, from a theoretical standpoint, it would be interesting to study the behaviour of EENN-AVCS as the number of exits goes to infinity. Implicit deep models [Chen et al., 2018, Bai et al., 2020] could be used to this end.

# 9   ACKNOWLEDGMENTS

We thank Alexander Timans, Rajeev Verma, and Mona Schirmer for helpful discussions. We are also grateful to the anonymous reviewers who helped us improve our work with their constructive feedback. MJ and EN are generously supported by the Bosch Center for Artificial Intelligence. SM acknowledges support by the IARPA WRIVA program, the National Science Foundation (NSF) under the NSF CAREER Award 2047418; NSF Grants 2003237 and 2007719, the Department of Energy, Office of Science under grant DE-SC0022331, as well as gifts from Disney and Qualcomm.

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

# A ADDITIONAL RESULTS

## A.1 SPEEDING UP CONVERGENCE OF EENN-AVCS

In our original formulation in Section 3, we draw a single sample of the weighs $\boldsymbol{W}_t$ (or predictive distribution $\boldsymbol{\mu}_t$ in the case of classification) at each exit. This invariably leads to large prediction intervals/sets at the initial exits - a phenomenon analogous to AVCSs being large for the initial few observed data points in the conventional data streaming scenario [Howard et al., 2020]. In this section, we explore two distinct approaches to mitigate this issue, aiming to attain more efficient confidence estimates right from the initial exits.

In the first approach, we simply take multiple samples $S_t > 1$ at each exit. Consequently, the predictive likelihood ratio for a given test point $\boldsymbol{x}^*$ takes the following form:

$$R_t(y) \;:=\; \prod_{l=1}^{t} \prod_{s=1}^{S_l} \frac{p_l(y|\boldsymbol{x}^*, \mathcal{D})}{p(y|\boldsymbol{x}^*, \boldsymbol{W}_l^{(s)})}, \;\; \boldsymbol{W}_l^{(s)} \sim p(\mathbf{W}_l|\mathcal{D}) \,.$$

We term this approach *Multiple-Samples AVCS*. As an alternative, we construct multiple AVCSs $\{C_t^{(s)}\}_{s=1}^{S_t}$ based on a single sample in parallel. At each exit, we then consider their intersection $C_t^\cap = \bigcap_{s=1}^{S_t} C_t^{(s)}$ and pass it on to the next exit. We refer to this method as *Parallel AVCS*.

We present the results for both approaches in Figure 7 using synthetic datasets from Section 7.1. While both methods yield more efficient, i.e., smaller, intervals in the initial exits (*top* row), it is interesting to observe that the *Multiple-Samples* approach leads to a much faster decay in marginal coverage compared to the *Parallel* one (see *bottom* row). We attribute this to the fact that by sampling multiple samples within a single confidence sequence at each exit, we are essentially 'committing' more to our approximation of the updated posterior (c.f., Eq. (3)), which results in larger coverage violations. Hence, we recommend using the *Parallel* approach when attempting to speed up the convergence of our EENN-AVCS . Nonetheless, we acknowledge that this area warrants further investigation, and we consider this an important direction for future work.

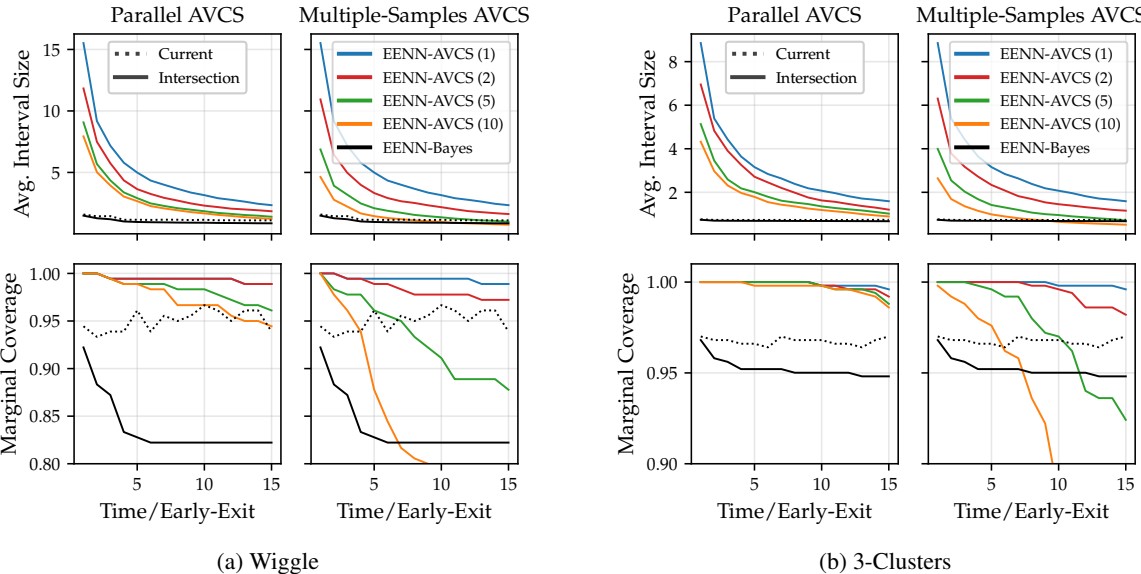

Figure 7: Average interval size and marginal coverage for regression synthetic datasets. While both of the considered approaches yield more efficient intervals (*top* row), the *Parallel* method is better at preserving high marginal coverage (*bottom* row). AVCS($S$) denotes a confidence sequence based on $S$ samples at each exit in the case of *Multiple-Samples*, and the sequence based on $S$ parallel ones in the case of *Parallel*.

# B SUPPORTING DERIVATIONS

## B.1 PROOF OF PROPOSITION 1

The proof can be divided into two steps. In the first step, we demonstrate that the predictive-likelihood ratio $R_t^*(y)$ in (2) is a non-negative martingale when evaluated at the true value $y^*$, with an initial value of one. In the second step, we utilize Ville's inequality to construct AVCS. Throughout this process, we closely adhere to the proof technique outlined in Waudby-Smith and Ramdas [2020] (refer to Appendix B.1 in that work).

We begin the first step by showing that the expectation of the predictive-likelihood ratio evaluated at $y^*$ remains constant over time:

$$\mathbb{E}_{\mathbf{W}_{t+1}}[R_{t+1}^*(y^*) \mid \mathbf{W}_1, \dots, \mathbf{W}_t] =$$

$$\int R_{t+1}^*(y^*) \, p(\mathbf{W}_{t+1}|\mathcal{D} \cup (\boldsymbol{x}^*, y^*)) \, d\mathbf{W}_{t+1} \overset{(i)}{=}$$

$$\int R_{t+1}^*(y^*) \, \frac{p(y^*|\boldsymbol{x}^*, \mathbf{W}_{t+1})p(\mathbf{W}_{t+1}|\mathcal{D})}{p_{t+1}(y^*|\boldsymbol{x}^*, \mathcal{D})} \, d\mathbf{W}_{t+1} =$$

$$\int \prod_{l=1}^{t+1} \frac{p_l(y^*|\boldsymbol{x}^*, \mathcal{D})}{p(y^*|\boldsymbol{x}^*, \mathbf{W}_l)} \, \frac{p(y^*|\boldsymbol{x}^*, \mathbf{W}_{t+1})p(\mathbf{W}_{t+1}|\mathcal{D})}{p_{t+1}(y^*|\boldsymbol{x}^*, \mathcal{D})} \, d\mathbf{W}_{t+1} =$$

$$\int \underbrace{\prod_{l=1}^{t} \frac{p_l(y^*|\boldsymbol{x}^*, \mathcal{D})}{p(y^*|\boldsymbol{x}^*, \mathbf{W}_l)}}_{R_t^*(y^*)} \, \frac{\cancel{p_{t+1}(y^*|\boldsymbol{x}^*, \mathcal{D})}}{\cancel{p(y^*|\boldsymbol{x}^*, \mathbf{W}_{t+1})}} \, \frac{\cancel{p(y^*|\boldsymbol{x}^*, \mathbf{W}_{t+1})} p(\mathbf{W}_{t+1}|\mathcal{D})}{\cancel{p_{t+1}(y^*|\boldsymbol{x}^*, \mathcal{D})}} \, d\mathbf{W}_{t+1} =$$

$$\int R_t^*(y^*) \, p(\mathbf{W}_{t+1}|\mathcal{D}) \, d\mathbf{W}_{t+1} =$$

$$R_t^*(y^*) \int p(\mathbf{W}_{t+1}|\mathcal{D}) \, d\mathbf{W}_{t+1} =$$

$$R_t^*(y^*) \, ,$$

where the step $(i)$ follows from the (sequential) Bayesian updating of the current posterior $p(\mathbf{W}_{t+1}|\mathcal{D})$ based on the new data-point $(\boldsymbol{x}^*, y^*)$.

To show that initial value is equal to one, we proceed similarly:

$$\mathbb{E}_{\mathbf{W}_1}[R_1^*(y^*)] =$$

$$\int R_1^*(y^*) \, p(\mathbf{W}_1|\mathcal{D} \cup (\boldsymbol{x}^*, y^*)) \, d\mathbf{W}_1 =$$

$$\int R_1^*(y^*) \, \frac{p(y^*|\boldsymbol{x}^*, \mathbf{W}_1)p(\mathbf{W}_1|\mathcal{D})}{p_1(y^*|\boldsymbol{x}^*, \mathcal{D})} \, d\mathbf{W}_1 =$$

$$\int p(\mathbf{W}_1|\mathcal{D}) \, d\mathbf{W}_1 = 1 =: R_0^* \, .$$

In the second step, we make use of Ville's inequality, which provides a bound on the probability that a non-negative supermartingale exceeds a threshold $\beta > 0$.

$$\mathbb{P}\left(\exists t : R_t^*(y^*) \geq \beta\right) \leq \mathbb{E}[R_0^*(y^*)] \, / \, \beta \, .$$

Since every martingale is also a supermartingale, Ville's inequality is applicable in our case. Then, for a particular threshold $\alpha \in (0, 1)$ and since we have a constant initial value (one), Ville's inequality implies: $\mathbb{P}(\exists t : R_t^*(y^*) \geq 1/\alpha) \leq \alpha$. If we define the sequence of sets as $C_t^* := \{y \in \mathcal{Y} \mid R_t^*(y) \leq 1/\alpha\}$, their validity can be shown as

$$\mathbb{P}(\forall t, y^* \in C_t^*) = \mathbb{P}(\forall t, R_t^*(y^*) \leq 1/\alpha) =$$
$$1 - \mathbb{P}(\exists t : R_t^*(y^*) \geq 1/\alpha) \geq 1 - \alpha \, ,$$

which concludes the proof.

## B.2  PROOF OF PROPOSITION 2

We first note that due to $C_t^*$ being a valid $(1-\alpha)$ confidence sequence, we have

$$P(\exists l \in [t], y^* \notin C_l^*) \leq P(\exists l \in [T], y^* \notin C_l^*) \leq \alpha \,, \tag{5}$$

where we adopt the notation $[t] := \{1, \ldots, t\}$ for brevity. Additionaly we observe that randomness in $P(\exists l \in [t], y^* \notin C_l)$ and $P(\exists l \in [t], y^* \notin C_l^*)$ comes from $p(\mathbf{W}_1, \ldots, \mathbf{W}_t | \mathcal{D})$ and $p(\mathbf{W}_1, \ldots, \mathbf{W}_t | \mathcal{D}_*)$, respectively. Hence, we can use total variation distance (TV) to upper bound the difference

$$P(\exists l \in [t], y^* \notin C_l) - P(\exists l \in [t], y^* \notin C_l^*) \leq$$
$$\left| P(\exists l \in [t], y^* \notin C_l) - P(\exists l \in [t], y^* \notin C_l^*) \right| \leq$$
$$TV\big(p(\mathbf{W}_1, \ldots, \mathbf{W}_t | \mathcal{D}),\, p(\mathbf{W}_1, \ldots, \mathbf{W}_t | \mathcal{D}_*)\big) \,.$$

Next, we apply Bretangnolle and Huber inequality [Bretagnolle and Huber, 1979] to upper bound the TV distance in terms of KL divergence and use the fact that weights at different exits are independent which gives rise to a factorized joint distribution

$$TV\big(p(\mathbf{W}_1, \ldots, \mathbf{W}_t | \mathcal{D}),\, p(\mathbf{W}_1, \ldots, \mathbf{W}_t | \mathcal{D}_*)\big) \leq$$
$$\sqrt{1 - e^{-KL\big(p(\mathbf{W}_1, \ldots, \mathbf{W}_t | \mathcal{D}),\, p(\mathbf{W}_1, \ldots, \mathbf{W}_t | \mathcal{D}_*)\big)}} \leq$$
$$\sqrt{1 - e^{-\sum_{l=1}^{t} KL\big(p(\mathbf{W}_l | \mathcal{D}),\, p(\mathbf{W}_l | \mathcal{D}_*)\big)}}$$

Rearranging the terms and using (5), the proposition follows

$$P(\exists l \in [t], y^* \notin C_l) \leq$$
$$P(\exists l \in [t], y^* \notin C_l^*) + \sqrt{1 - e^{-\sum_{l=1}^{t} KL_l}} \leq$$
$$\alpha + \sqrt{1 - e^{-\sum_{l=1}^{t} KL_l}}$$

where $KL_l := KL\big(p(\mathbf{W}_l | \mathcal{D}),\, p(\mathbf{W}_l | \mathcal{D}_*)\big)$.

## B.3  BAYESIAN LINEAR REGRESSION

In Section 4, we define the predictive model at the $t$th exit as a linear model $f(\boldsymbol{x}; \boldsymbol{W}_t, \boldsymbol{U}_{1:t}) = h(\boldsymbol{x}; \boldsymbol{U}_{1:t})^T \boldsymbol{W}_t$. For notational brevity, we omit $\boldsymbol{U}_{1:t}$ and denote $h(\boldsymbol{x}; \boldsymbol{U}_{1:t})$ as $h_t(\boldsymbol{x})$ in this section. Additionally, let $\boldsymbol{y} = [y_1, \ldots, y_N]^T \in \mathbb{R}^N$ and $\boldsymbol{H}_t = [h_t(\boldsymbol{x}_1), \ldots, h_t(\boldsymbol{x}_N)]^T \in \mathbb{R}^{N \times H}$ represent a concatenation of training labels and (deep) features, respectively. Assuming a Gaussian likelihood $\mathcal{N}\big(\mathrm{y}; h_t(\boldsymbol{x})^T \boldsymbol{W}_t, \sigma_t^2\big)$ and a prior $\mathcal{N}\big(\boldsymbol{W}_t; \boldsymbol{0}, \sigma_{w,t}^2 \mathbb{I}_H\big)$, the posterior over weights $\mathbf{W}_t$ has the following form [Bishop and Nasrabadi, 2006]:

$$p(\mathbf{W}_t | \mathcal{D}) = \mathcal{N}\big(\mathbf{W}_t; \bar{\boldsymbol{\mu}}_t, \bar{\boldsymbol{\Sigma}}_t\big) \,,$$
$$\bar{\boldsymbol{\mu}}_t = \frac{1}{\sigma_t^2} \bar{\boldsymbol{\Sigma}}_t \boldsymbol{H}_t^T \boldsymbol{y} \,,$$
$$\bar{\boldsymbol{\Sigma}}_t^{-1} = \frac{1}{\sigma_t^2} \boldsymbol{H}_t^T \boldsymbol{H}_t + \frac{1}{\sigma_{w,t}^2} \mathbb{I}_H \,.$$

Similarly, for a new test point $\boldsymbol{x}^*$, the posterior predictive can be obtained in a closed-form:

$$p_t(\mathrm{y} | \boldsymbol{x}^*, \mathcal{D}) = \mathcal{N}\big(\mathrm{y}; h_t(\boldsymbol{x}^*)^T \bar{\boldsymbol{\mu}}_t, h_t(\boldsymbol{x}^*)^T \bar{\boldsymbol{\Sigma}}_t h_t(\boldsymbol{x}^*) + \sigma_t^2\big) \,.$$

For the exact derivation of both distributions above, we refer the interested reader to the Section 3.3 in Bishop and Nasrabadi [2006].

## B.4 SOLVING FOR INTERVAL ENDPOINTS

Due to the assumed Bayesian linear regression model at each exit $t$, $\log R_t$ is a convex quadratic function in $y$:

$$\log R_t(y) =$$
$$\sum_{l=1}^{t} \log p_l(y|\boldsymbol{x}^*, \mathcal{D}) - \log p(y|\boldsymbol{x}^*, \boldsymbol{W}_l) =$$
$$\alpha_t(\boldsymbol{x}^*) \cdot y^2 + \beta_t(\boldsymbol{x}^*, \boldsymbol{W}_{1:t}) \cdot y + \gamma_t(\boldsymbol{x}^*, \boldsymbol{W}_{1:t}) \,.$$

Coefficients have the following form:

$$\alpha_t(\boldsymbol{x}^*) = \frac{1}{2} \sum_{l=1}^{t} \left( \frac{1}{\sigma_l^2} - \frac{1}{v_{*,l} + \sigma_l^2} \right),$$

$$\beta_t(\boldsymbol{x}^*, \boldsymbol{W}_{1:t}) = \sum_{l=1}^{t} \frac{h_l(\boldsymbol{x}^*)^T \bar{\boldsymbol{\mu}}_l}{v_l^* + \sigma_l^2} - \frac{h_l(\boldsymbol{x}^*)^T \boldsymbol{W}_l}{\sigma_l^2},$$

$$\gamma_t(\boldsymbol{x}^*, \boldsymbol{W}_{1:t}) =$$
$$\frac{1}{2} \sum_{l=1}^{t} \left( \frac{(h_l(\boldsymbol{x}^*)^T \boldsymbol{W}_l)^2}{\sigma_l^2} - \frac{(h_l(\boldsymbol{x}^*)^T \bar{\boldsymbol{\mu}}_l)^2}{v_l^* + \sigma_l^2} + \log \frac{\sigma_l^2}{v_l^* + \sigma_l^2} \right)$$

where $v_l^* := h_l(\boldsymbol{x}^*)^T \bar{\boldsymbol{\Sigma}}_l h_l(\boldsymbol{x}^*)$, and we provide expressions for $h_l, \bar{\boldsymbol{\mu}}_l, \bar{\boldsymbol{\Sigma}}_l$ in Appendix B.3. It is easy to show that $\alpha_t \geq 0$, from which the convexity follows.

To find AVCS $C_t = \{y \in \mathcal{Y} \mid R_t(y) \leq 1/\alpha\}$, we look for the roots of the equation $\log R_t(y) - \log(1/\alpha) = 0$. This yields an analytical expression for $C_t = [y_L^t, y_R^t]$ :

$$y_{L,R}^t = \frac{-\beta_t \pm \sqrt{\beta_t^2 - 4\alpha_t \tilde{\gamma}_t}}{2\alpha_t}$$

where $\tilde{\gamma}_t = \gamma_t + \log \alpha$. See Figure 8 for a concrete example of log-ratios.

## B.5 EPISTEMIC UNCERTAINTY AND KL DIVERGENCE

To compute the KL divergence between the posterior and update posterior in the Bayesian linear regression model (c.f. Appendix B.3), we first use the Bayes rule to rewrite the latter as:

$$p(\mathbf{W}_t|\mathcal{D}_*) = \frac{p(y^*|\boldsymbol{x}^*, \mathbf{W}_t)\, p(\mathbf{W}_t|\mathcal{D})}{p_t(y^*|\boldsymbol{x}^*, \mathcal{D})} \,.$$

Using the definition of the KL divergence together with the formulas for posterior predictive and posterior distributions from Appendix B.3, we proceed as

$$KL\big(p(\mathbf{W}_t|\mathcal{D}), p(\mathbf{W}_t|\mathcal{D}_*)\big) =$$
$$\mathbb{E}_{p(\mathbf{W}_t|\mathcal{D})} \left[ \log \frac{p(\mathbf{W}_t|\mathcal{D})}{p(\mathbf{W}_t|\mathcal{D}_*)} \right] =$$
$$\log p_t(y^*|\boldsymbol{x}^*, \mathcal{D}) - \mathbb{E}_{p(\mathbf{W}_t|\mathcal{D})} \big[ \log p(y^*|\boldsymbol{x}^*, \mathbf{W}_t) \big] =$$
$$0.5 \left( \log \big( \frac{\sigma_t^2}{\sigma_t^2 + v_t^*} \big) + \big( \frac{1}{\sigma_t^2 + v_t^*} - \frac{1}{\sigma_t^2} \big) r_*^2 + \frac{v_t^*}{\sigma_t^2} \right)$$

where $r_* = y^* - \bar{\boldsymbol{\mu}}_t^T h_t(\boldsymbol{x}^*)$ represents a residual, $v_* = h_t(\boldsymbol{x}^*)^T \bar{\boldsymbol{\Sigma}}_t h_t(\boldsymbol{x}^*)$ denotes epistemic uncertainty, and $\sigma = \sigma_{y,t}$. Based on the obtained expression, it is evident that a small $v^*$, implies small KL-divergence.

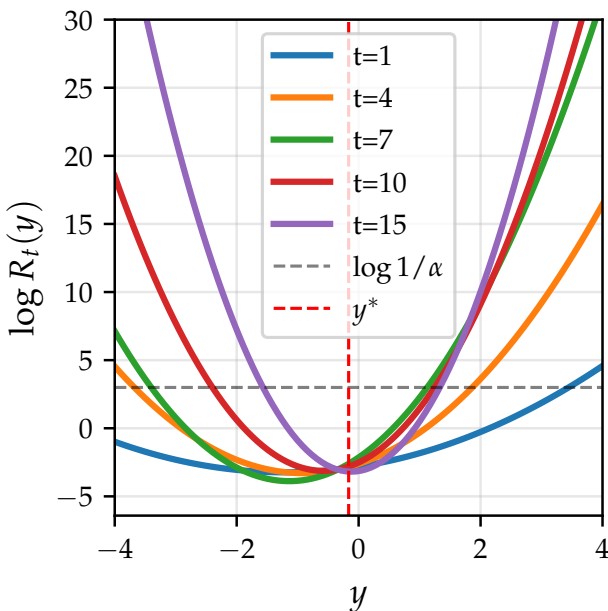

Figure 8: Plot of $\log R_t(y)$ at various exits $t$ for a randomly selected test data point $(\boldsymbol{x}^*, y^*)$ from the *3-clusters* dataset. As described in Appendix B.4, we observe that the log-ratios exhibit a quadratic shape, allowing for an analytical solution for the endpoints of prediction intervals $C_t$.

## C IMPLEMENTATION DETAILS

### C.1 SYNTHETIC DATA EXPERIMENTS

**Data Generation**  We closely follow data generation process from Antorán et al. [2020]. Specifically, for *wiggle* dataset we sample $N$ points from

$$y = \sin(\pi x) + 0.2\cos(4\pi x) - 0.3x + \epsilon$$

where $\epsilon \sim \mathcal{N}(0, 0.25)$ and $x \sim \mathcal{N}(5, 2, 5)$. For *3-clusters* dataset, we simulate data via

$$y = x - 0.1x^2 + \cos(x\pi/2)$$

where $\epsilon \sim \mathcal{N}(0, 0.25)$ and we sample $N/3$ points from $[-1, 0]$, $[1.5, 2.5]$ and $[4, 5]$, respectively. For both datasets, we sample a total of $N = 900$ points and allocate $80\%$ of the data for training, while the remaining $20\%$ constitutes the test dataset.

**Model Architecture**  Our EENN is composed of an input layer and $T = 15$ residual blocks. The residual blocks consist of a `Dense` layer (with $M = 20$ hidden units), followed by a `ReLU` activation and `BatchNorm` (with default `PyTorch` parameters). We attach an output layer at each residual block to facilitate early exiting.

**Training**  We train our EENN for 500 epochs using `SGD` with a learning rate of $1 \times 10^{-3}$, a momentum of 0.9, and a weight decay of $1 \times 10^{-4}$. For the loss function, we use the average mean-square error (`MSE`) across all exits.

## C.2 SEMANTIC TEXTUAL SIMILARITY EXPERIMENT

**Datasets**  We use the STS-B dataset, the only regression dataset in the GLUE benchmark [Wang et al., 2019], as well as the SICK dataset [Marelli et al., 2014]. The task is to measure the semantic similarity $y \in [0, 5]$ between the two input sentences. For STS-B, the training, development, and test datasets consist of 5.7K, 1.5K, and 1.4K datapoints, respectively. For SICK, , the training, development, and test datasets consist of 4.4K, 2.7K, and 2.7K datapoints, respectively.

**Model Architecture and Training**  For the model architecture and training we reuse the code from Zhou et al. [2020]. Specifically, we work with `ALBERT-large` which is a 24-layers transformer model. To facilitate early exiting, a regression head is attached after every transformer block.

**EENN-AVCS**  In the results presented in the main text, we construct a single ($S = 1$) AVCS at test time with $\alpha = 0.05$. To fit the Bayesian linear regression models (i.e., empirical Bayes) at every exit, we use the development set. Note that this contrasts with our experiments on the synthetic dataset (c.f., Section 7.1) where we utilized the

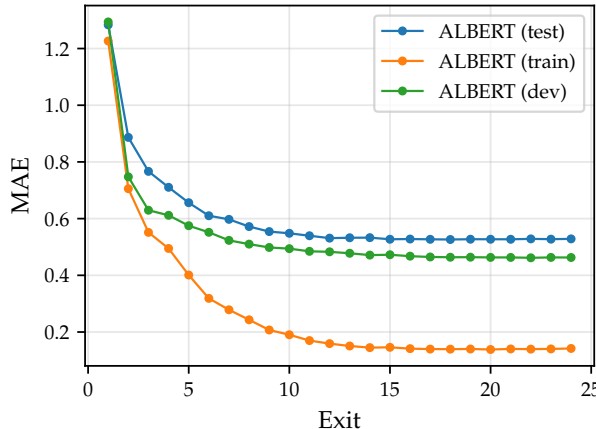

Figure 9: Mean Absolute Error (MAE) performance of the `ALBERT-large` model across different datasets: train, development (dev), and test. A large performance gap between the train and dev/test datasets is observed. Note that in our work, we reuse the exact model and training setup from previous approaches [Zhou et al., 2020].

training dataset for this purpose. We observed that when fitting the regression model on the training dataset for STS-B, the noise parameters $\hat{\sigma}_t$ get underestimated, resulting in a rapid decay of marginal coverage for both EENN-AVCS and EENN-Bayes . We attribute this to a distribution shift present in the STS-B dataset, which is evident based on the different performances (MAE) that the ALBERT model achieves on different datasets, as seen in Figure 9.

# D    EENN-AVCS ALGORITHM

Here, we outline in detail the implementation of our EENN-AVCS model. In Algorithm 1, we present EENN-AVCS for regression tasks. We start by fitting a Bayesian posterior model $p(\mathbf{W}_t|\mathcal{D})$ at every exit using the training data $\mathcal{D}$ (c.f. Appendix B.3). To estimate the observation noise $\hat{\sigma}_t$ at every exit, we perform empirical Bayes (type-II maximum likelihood). Then, for a given test point $\boldsymbol{x}^*$, we first sample the weights from the posterior and compute the epistemic uncertainty $v_t^*$ at every exit. Next, we use the obtained quantities to update the coefficients of the (logarithm of) predictive-likelihood ratio $R_t$ (c.f. Appendix B.4). To get the prediction interval at a given exit, we then solve the quadratic equation based on the updated coefficients from the previous step (c.f. Appendix B.4). Finally, we take the running intersection with the intervals obtained at the previous exits. In case the intersection results in an empty interval, we stop evaluating exits and label the given test point $\boldsymbol{x}^*$ as an out-of-distribution (OOD) example (c.f. *Detecting Violations of Posterior Stability* in Section 3).

In Algorithm 2, we present EENN-AVCS for classification tasks. To determine the concentration parameters $\boldsymbol{\alpha}_t$ of the Dirichlet distribution at each exit for a given test point $\boldsymbol{x}^*$, we apply a ReLU activation to the logits from the backbone EENN, retaining only the classes that "survive" the ReLU. We then sample from the Dirichlet distribution to obtain the denominator part of the predictive-likelihood ratio $R_t$ (refer to Section 5). For the numerator part of $R_t$, we calculate the (closed-form) posterior distribution using the concentration parameters at a specific exit. To create a predictive set at a given exit, we iterate over classes and include only those classes in the set for which the predictive-likelihood ratio $R_t$ is less than $1/\alpha_S$. Finally, as in the regression case, we consider the running intersection with all sets computed at previous exits. We label the test example $v^*$ as out-of-distribution (OOD) if the set collapses to an empty set.

---

**Algorithm 1:** EENN-AVCS Regression

**input** : Backbone EENN $\{h(\cdot|\boldsymbol{U}_{1:t})\}_{t=1}^T$, Regression models $\{p(\mathbf{W}_t|\mathcal{D}),\ \hat{\sigma}_t^2\}_{t=1}^T$, test datapoint $\boldsymbol{x}^*$, significance level $\alpha_S$

**output** : AVCS for $\boldsymbol{x}^*$

$C_0 = \mathcal{Y}$
$\alpha, \beta, \gamma = 0, 0, \log \alpha_S$
**for** $t = 1, ..., T$ **do**

$\quad \mathbf{W}_t \sim p(\mathbf{W}_t|\mathcal{D}) = \mathcal{N}(\mathbf{W}_t|\bar{\boldsymbol{\mu}}_t, \bar{\boldsymbol{\Sigma}}_t)$
$\quad v_t^* := h_t(\boldsymbol{x}^*)^T \bar{\boldsymbol{\Sigma}}_t h_t(\boldsymbol{x}^*)$
$\quad$ # update coefficients of $\log R_t(y)$
$\quad \alpha \mathrel{+}= \frac{1}{2}\left(\frac{1}{\hat{\sigma}_t^2} - \frac{1}{v_t^* + \hat{\sigma}_t^2}\right)$
$\quad \beta \mathrel{+}= \frac{h_t(\boldsymbol{x}^*)^T \bar{\boldsymbol{\mu}}_t}{v_t^* + \hat{\sigma}_t^2} - \frac{h_t(\boldsymbol{x}^*)^T \mathbf{W}_t}{\hat{\sigma}_t^2}$
$\quad \gamma \mathrel{+}= \frac{1}{2}\left(\frac{(h_t(\boldsymbol{x}^*)^T \mathbf{W}_t)^2}{\hat{\sigma}_t^2} - \frac{(h_t(\boldsymbol{x}^*)^T \bar{\boldsymbol{\mu}}_t)^2}{v_t^* + \hat{\sigma}_t^2} + \log\frac{\hat{\sigma}_t^2}{v_t^* + \hat{\sigma}_t^2}\right)$

$\quad$ # find the roots of quadratic equation
$\quad y_{L,R}^t = \frac{-\beta \pm \sqrt{\beta^2 - 4\alpha\gamma}}{2\alpha}$
$\quad C_t = C_{t-1} \cap [y_L^t, y_R^t]$
$\quad$ **if** $C_t = \emptyset$ **then**
$\quad\quad$ | **return** $\emptyset$ # OOD
**return** $\{C_t\}_{t=1}^T$

---

**Algorithm 2:** EENN-AVCS Classification

**input** : Backbone EENN $\{f(\cdot|\boldsymbol{U}_{1:t}, \boldsymbol{W}_t)\}_{t=1}^T$, ReLU thresholds $\{\tau_t\}_{t=1}^T$, test datapoint $\boldsymbol{x}^*$, significance level $\alpha_S$

**output** : AVCS for $\boldsymbol{x}^*$

$C_0 = \mathcal{Y}$
$R = [1, ..., 1]$
**for** $t = 1, ..., T$ **do**

$\quad$ # get concentration parameters, only keep classes that "survive" ReLU
$\quad \boldsymbol{\alpha}_t = \texttt{ReLU}(f(\boldsymbol{x}^*|\boldsymbol{U}_{1:t}, \boldsymbol{W}_t), \tau_t)$
$\quad \tilde{\boldsymbol{\alpha}}_t = \boldsymbol{\alpha}_t[\boldsymbol{\alpha}_t > 0]$
$\quad \boldsymbol{\pi}_t \sim \texttt{Dir}(\tilde{\boldsymbol{\alpha}}_t)$
$\quad S_t = \sum_k \alpha_{t,k}$
$\quad C_t = [\ ]$
$\quad$ # update the predictive-likelihood ratio
$\quad$ **for** $k = 1, \ldots, K$ **do**
$\quad\quad$ **if** $\alpha_{t,k} > 0$ **then**
$\quad\quad\quad$ | $R[k] \mathrel{*}= \frac{\alpha_{t,k}/S_t}{\pi_{t,k}}$
$\quad\quad$ **else**
$\quad\quad\quad$ | $R[k] = \infty$
$\quad\quad$ **if** $R[k] \leq \frac{1}{\alpha_S}$ **then**
$\quad\quad\quad$ | $C_t.\texttt{append}(k)$
$\quad C_t = C_t \cap C_{t-1}$
$\quad$ **if** $C_t = \emptyset$ **then**
$\quad\quad$ | **return** $\emptyset$ # OOD
**return** $\{C_t\}_{t=1}^T$

---

