# OpenReview forum: "Early-Exit Neural Networks with Nested Prediction Sets"
_auai.org/UAI/2024/Conference — UAI 2024 poster_

### Official Review · Reviewer_rfRd · 2024-03-19

**Q2-1 Originality-Novelty:** 3
**Q2-2 Correctness-Technical Quality:** 2
**Q2-5 Clarity Of Writing:** 4

**Q1 Summary And Contributions:**

This work studies the important problem of reliable uncertainty quantification of predictions from earl-exit neural networks (EENNs) by computing predictive confidence intervals (or sets) around the predictions at each exit, with the aim of computing intervals that are *consistent* (i.e., the interval $C_t$ at exit $t$ is a subset of every previous exit’s interval) and *anytime-valid* (i.e., for all possible exit times $t$, $C_t$ is a valid $1-\alpha$ confidence interval). The approach proposed is to leverage ideas from anytime-valid confidence sequences for EENNs, with formulations for both regression and classification tasks. Thorough experiments are reported on a semantic textual similarity regression task and an image classification task, with reasonable baselines (i.e., Bayesian and conformal inference methods).

**Q2-3 Extent To Which Claims Are Supported By Evidence:**

2: Fair: the main claims are somewhat supported by evidence (but the experimental evaluation may be weak, or does not match entirely with the claims, important baselines may be missing, proofs contain important ideas but lack rigor, algorithmic details are only discussed superficially, references are imprecise, assumptions are not sufficiently motivated or explicated, etc.).

**Q2-4 Reproducibility:**

3: Good: key resources (e.g. proofs, code, data) are available and key details (e.g. proofs, experimental setup) are sufficiently well-described for competent researchers to confidently reproduce the main results.

**Q3 Main Strengths:**

**1. Originality and novelty:** The motivation for the paper is strong, as EENNs are clearly an application where reliable uncertainty quantification is of critical importance, for instance in common applications where there are constraints on the amounts of available computational resources. The proposed use of anytime-valid confidence sequences for this problem is very reasonable and seems to be a novel contribution that would be of broad interest, assuming that the developed methods are technically sound.

**2. Reproducibility:** The paper seems to have strong reproducibility, for example with adequate experimental details and code made available.

**3. Clarity of writing:** Overall the paper was very clearly written, and I sincerely enjoyed reading it.

**Q4 Main Weakness:**

[Post-Rebuttal Note: See official comment below regarding updates on this point.]

**1. Correctness-technical quality / Extent To Which Claims Are Supported By Evidence:** Unfortunately, there appears to be an issue with the first step of the proof for Proposition 1 (a main result) requiring an assumption that is not stated as an assumption for that proposition or its proof, which prevents me from recommending this paper for acceptance in its current form. As I explain in the “Detailed Comments to the Authors”, this seems to be more than a “typo,” because the assumption is introduced in the main paper as precisely what makes the proposed method "realizable" rather than "idealized", but then the same assumption appears to be implicitly required by the proof for the "idealized" method's guarantee given in Proposition 1, meaning that simply stating this assumption as required for the "idealized" construction seems like it would erase any meaningful difference between the "idealized" and "realizable" categories given. As I describe in the Detailed Comments, I would be open to considering either clarifications from the authors if I am somehow misunderstanding something or revisions to fix the error if I am not.

**Q5 Detailed Comments To The Authors:**

[Post-Rebuttal Note: See official comment below regarding updates on this first point.]

**1. Detailed comments on the potential issue with the first step of the proof for Proposition 1, given in Appendix C.1:**

*The Context for the Potential Issue:*

 - *”Idealized Construction”:* Proposition 1 is presented in part of Section 3 titled “Idealized Construction”, as Proposition 1 is intended to give the guarantee that would be obtained in an idealized (but not practical) world where one has access to the test point $(x^*, y^*)$. In particular, the Idealized Construction relies on fitting the posterior $p(W_l | \mathcal{D} \cup (x^*, y^*))$ on both the training data $\mathcal{D}$ *and* the test point $(x^*, y^*)$.

- *”Realizable Relaxation”* The idealized construction is *in distinct contrast to* the subsequently presented “Realizable Relaxation,” which is defined by assuming that one could simply ignore the reliance on the test point $(x^*, y^*)$ in the Idealized Construction. This is stated in assumption (3), which is saying that the posterior estimate of the parameters $W_l$ is approximately stable with respect to the potential dependency on the test point $(x^*, y^*)$, that is: $p(W_{t} | \mathcal{D} \cup (x^*, y^*)) \approx p(W_{t} | \mathcal{D})$. In other words, the main paper presents assumption (3) as exactly what distinguishes the “Realizable Relaxation” from the “Idealized Construction” and what makes it “realizable”: i.e., by assuming you can roughly ignore the dependency on the test point $(x^*, y^*)$.

*The Issue:* With this context established--i.e., with assumption (3) being precisely what distinguishes the “Realizable Relaxation” from the “Idealized Construction” by making it “realizable”--the issue becomes apparent in a close reading of the first part of the proof for Proposition 1 given in Appendix C.1. That is, this proof for the “Idealized Construction” appears to rely on an even stronger assumption than assumption (3) (which shouldn’t be the case if the only purpose of introducing this assumption in the main paper was for “realizability”): namely, the proof seems to require assumption (3) where the “approximately equal” relation is replaced with equality: $p(W_{t} | \mathcal{D} \cup (x^*, y^*)) = p(W_{t} | \mathcal{D})$ (the time index is arbitrary here, it seems).

Specifically, the justification for step (i) in the first step of the proof for Proposition 1 presented in Appendix C.1 uses the substitution $p(W_{t+1} | x^*, \mathcal{D}) = p(W_{t+1} | \mathcal{D})$, which is essentially saying that one can ignore the dependency on the test point, and would follow if the proof assumed the stronger version of assumption (3) mentioned. The necessity of (the stronger version of) assumption (3) (with an equality relation) is even easier to see if one ignores the result of step (i); i.e., if one compares the second line in the proof to the fourth line, the only difference is the substitution $p(W_{t} | \mathcal{D} \cup (x^*, y^*)) = p(W_{t} | \mathcal{D})$, which is exactly the stronger version of assumption (3) that I have described.

Due to this key assumption that appears to be necessary for proving Proposition 1 but is not stated as an assumption for that proposition or for its proof, I unfortunately do not feel able to recommend this paper for acceptance in its current form. Moreover, this unmentioned assumption for the proof of Proposition 1 seems to be more than a “typo” because it is introduced as precisely what defines the “Realizable Relaxation” as distinct from the “Idealized Construction”, as I have described, despite apparently being necessary for the proof of the “Idealized Construction”.

*Potential fixes:* However, I would consider updating my recommendation if the authors are able to revise the manuscript to address this issue (or if I am somehow misunderstanding something and they are able to clarify this). Specifically, it seems like the authors could potentially do one of the following:

  - (a) State (the stronger version of) assumption (3) as an assumption for Proposition 1 and for its proof. This seems the least work, but may raise further confusion about why an assumption of the “Realizable Relaxation” is apparently needed for the “Idealized Construction”.

  - (b) Change how the paper defines the predictive-likelihood ratio (2) in the “Idealized Construction” so that ultimately in the proof the substitution involving assumption (3) is not required. This would thereby address confusions that could arise in (a), but would require determining whether there is a different oracle/idealized construction that could serve as better reference.

  - (c) Restructure the relevant parts of the paper so that it no longer depends on the distinction between the “Idealized Construction” and the “Realizable Relaxation”. This may be the best course on balance, e.g., by simply stating (the strong version of) assumption (3) as an assumption made in the work overall (as it seems necessary for both theory and practice in the current formulation), and then by removing the current distinction between the “Idealized Construction” and the “Realizable Relaxation”.

**2. A further comment on assumption (3):** In future work (if not in a revision of this work), it could be beneficial to more precisely define what is meant by the “approximately equal to” in assumption (3)’s statement $p(W_{t} | \mathcal{D} \cup (x^*, y^*)) \approx p(W_{t} | \mathcal{D})$. For instance, if the data are all i.i.d., this seems to correspond to a “leave-one-out stability” assumption, which can be written more precisely as something like “with high probability, the difference between the posteriors is less than some $\epsilon$; for an example can see Section 5 in following reference: Barber, R. F., Candes, E. J., Ramdas, A., & Tibshirani, R. J. (2021). Predictive inference with the jackknife+.

**3. Notational clarity, depending on $D$ versus $W_l$:** I sometimes found the notation a bit confusing when going between conditioning the posterior on $D$ versus on $W_l$. It seems that the dependency on $D$ or $D*$ is always via some learned parameters $W_l$; if this is the case it could be helpful to sometimes simply write the parameters as a function of the data, i.e., $W_l(D)$ to more easily disambiguate between $W_l(D)$ and $W_l(D*)$.

**4. Section 5 description of Bayesian baselines:** In Section 5, instead of “depart from the Bayesian predictive model and instead utilize Dirichplet Prior Networks…” should this instead say something like “depart from *Gaussian assumptions*”? Because what is described still seems Bayesian, just not Bayesian in the common sense of using Gaussian priors and likelihoods for conjugacy.

**5. Section 6:** The target coverage for the experiments (i.e., what is $1-\alpha$) could be stated earlier in the section and in the figure captions for clarity. Would also help if could add a dotted line at the target coverage level in the relevant subplots.

**Q9 Complying With Reviewing Instructions:**

Yes

---

> ### Author Rebuttal · Authors · 2024-04-01
>
> We thank the reviewer for a comprehensive and very insightful review. We sincerely appreciate your efforts.
>
> **Q1**: Unfortunately, it seems like there has been some misunderstanding around our main proof. To resolve this, we outline the proof here and expand the predictive-likelihood ratio $R_t^*$ (see eq. (2) in the manuscript) for clarity:
>
> \begin{align*}
> \mathbb{E}\_{\mathbf{W}\_{t+1}}[R\_{t+1}^*(y^*) \\: | \\: \mathbf{W}\_1, \ldots, \mathbf{W}\_t] = \newline \int R\_{t+1}^*(y^*) \\: p (\mathbf{W}\_{t+1} | \mathcal{D} \cup (\boldsymbol{x}^*, y^*)) \\: d\mathbf{W}\_{t+1} \stackrel{(i)}{=} \newline
> \int R\_{t+1}^*(y^*) \\: \frac{p(y^* | \boldsymbol{x}^*, \mathbf{W}\_{t+1}) p(\mathbf{W}\_{t+1} | \mathcal{D})}{p\_{t+1}(y^*| \boldsymbol{x}^*,  \mathcal{D})} \\: d\mathbf{W}\_{t+1}  = \newline \int \prod\_{l=1}^{t+1} \frac{p\_{l}(y^*| \boldsymbol{x}^*,  \mathcal{D})}{p(y^* | \boldsymbol{x}^*, \mathbf{W}\_{l})}  \\: \frac{p(y^* | \boldsymbol{x}^*, \mathbf{W}\_{t+1}) p(\mathbf{W}\_{t+1} | \mathcal{D})}{p\_{t+1}(y^*| \boldsymbol{x}^*,  \mathcal{D})} d\mathbf{W}\_{t+1}= \newline \int \underbrace{\prod\_{l=1}^{t} \frac{p\_{l}(y^*| \boldsymbol{x}^*,  \mathcal{D})}{p(y^* | \boldsymbol{x}^*, \mathbf{W}\_{l})}}\_{=R^*\_t(y^*)} \cdot \frac{\cancel{p\_{t + 1}(y^*| \boldsymbol{x}^*,  \mathcal{D})}}{\cancel{p(y^* | \boldsymbol{x}^*, \mathbf{W}\_{t + 1})}} \cdot \frac{\cancel{p(y^* | \boldsymbol{x}^*, \mathbf{W}\_{t+1})} p(\mathbf{W}\_{t+1} | \mathcal{D})}{\cancel{p\_{t+1}(y^*| \boldsymbol{x}^*,  \mathcal{D})}} d\mathbf{W}\_{t+1} = \newline
> \int R\_{t}^*(y^*) \\: p(\mathbf{W}\_{t+1} | \mathcal{D}) \\: d\mathbf{W}\_{t+1} = \newline
> R\_{t}^*(y^*) \underbrace{\int \\: p(\mathbf{W}\_{t+1} |  \mathcal{D}) \\: d\mathbf{W}\_{t+1}}\_{=1} =  \newline
> R\_{t}^*(y^*)
> \end{align*}
>
> We will add this additional step to the proof in the Appendix C.1 in the camera-ready version. Hopefully, this makes it clear that the assumption $p(\mathbf{W}\_{t+1} | \mathcal{D} \cup (\boldsymbol{x}^*, y^*)) = p(\mathbf{W}\_{t+1} | \mathcal{D})$  is **not** needed for the proof. Rather, the martingale property follows due to our construction of predictive-likelihood ratio such that the respective distributions cancel as necessary.
>
> We also note that the step $(i)$ follows from the [(sequential) Bayesian updating](https://stats.stackexchange.com/questions/181934/sequential-update-of-bayesian) of the existing posterior $p(\mathbf{W}\_{t+1} | \mathcal{D})$ based on the new data-point $(\boldsymbol{x}^*, y^*)$:
>
> $$
> p (\mathbf{W}\_{t+1} | \mathcal{D} \cup (\boldsymbol{x}^*, y^*)) = \frac{p( y^* | \boldsymbol{x}^*, \mathbf{W}\_{t+1}) p(\mathbf{W}\_{t+1} | \mathcal{D})}{p\_{t+1}( y^*| \boldsymbol{x}^*,\mathcal{D})}
> $$
>
> where $p( y^* | \boldsymbol{x}^*, \mathbf{W}\_{t+1})$ is the likelihood and $p(\mathbf{W}\_{t+1} | \mathcal{D})$ is the posterior based only on the data $\mathcal{D}$ (it serves the role of the prior in the updating rule). The evidence term corresponds to the posterior predictive:
>
> $$
>  \int p( y^* | \boldsymbol{x}^*, \mathbf{W}\_{t+1}) p(\mathbf{W}\_{t+1} | \mathcal{D}) d\mathbf{W}_{t+1} = p\_{t+1}( y^*| \boldsymbol{x}^*,\mathcal{D})
> $$
>
> In the current version of the manuscript, we state that $(i)$ follows from the Bayes rule and additional assumptions of $p( y^* |\mathcal{D}, \boldsymbol{x}^*, \mathbf{W}\_{t+1}) = p( y^* |  \boldsymbol{x}^*, \mathbf{W}\_{t+1})$  and $p(\mathbf{W}\_{t+1} | \mathcal{D}, \boldsymbol{x}^*) = p(\mathbf{W}\_{t+1} | \mathcal{D})$. Following your review, we see how this might have led to confusion. We will change the argumentation in the camera-ready to follow the aforementioned Bayesian updating, and we thank you for raising our awareness about the potential source of confusion here.
>
> Lastly, we would politely point out, that the lines 2 and 4 in the current proof of Proposition 1 (see Appendix C.1) do not differ only in the posterior, i.e.  $p(\mathbf{W}\_{t+1} | \mathcal{D} \cup (\boldsymbol{x}^*, y^*))$ vs. $p(\mathbf{W}\_{t+1} | \mathcal{D})$, but crucially also in the time index of the predictive-likelihood ratio, i.e. $R\_{\textcolor{red}{t+1}}^*(y^*)$ vs. $R\_{\textcolor{red}{t}}^*(y^*)$.
>
> If there are any other open points about the proof of Prop. 1, please let us know, we are happy to engage in further discussion. Otherwise, if you feel we have sufficiently addressed your main concerns, we would appreciate if you would consider increasing your score.
>
> **Q2**: We make initial step towards quantifying the approximation more precisely in Appendix (see sections B.2 and C.5) where we look at the KL divergence between the two posteriors, but we appreciate your suggestion on jackknife+, will look closer into it!
>
> **Q4**: What we meant in Section 5 is that we move away from "being Bayesian" about the weights, to being Bayesian about model's outputs. Will make this more explicit in the camera-ready.
>
> **Q5**: Noted, will add an earlier mention of the target coverage and add a dotted line to the plots, thanks!

---

### Official Review · Reviewer_EE6M · 2024-03-21

**Q2-1 Originality-Novelty:** 3
**Q2-2 Correctness-Technical Quality:** 3
**Q2-5 Clarity Of Writing:** 2

**Q1 Summary And Contributions:**

In this paper, the authors present anytime valid confidence sequences based approach for early exit neural networks. Early exit neural networks can suffer from the problem when the predictions coming from each exit are inconsistent with the prior exits i.e. a prediction that was not included in the prior exit comes up in the subsequent exist, thus creating "consistency" issues. The authors extend anytime valid confidence sequences to the confidence intervals coming out of each exit by treating exit parameters as coming from a stream. The authors describe challenges in adapting this to regression and classification domains for deep neural networks, and provide an approach Bayesian inference on exit layers. The empirical results show interesting results against the baselines involving vanilla Bayesian inference and conformal prediction for early exit neural networks.

**Q2-3 Extent To Which Claims Are Supported By Evidence:**

2: Fair: the main claims are somewhat supported by evidence (but the experimental evaluation may be weak, or does not match entirely with the claims, important baselines may be missing, proofs contain important ideas but lack rigor, algorithmic details are only discussed superficially, references are imprecise, assumptions are not sufficiently motivated or explicated, etc.).

**Q2-4 Reproducibility:**

2: Fair: key resources (e.g. proofs, code, data) are unavailable but key details (e.g. proof sketches, experimental setup) are sufficiently well-described for an expert to confidently reproduce the main results.

**Q3 Main Strengths:**

- The adaptation of AVCS for early exit neural networks seems novel in my opinion, and the challenges that the authors have described and  their solutions are indicative of non-trivial advancements.

- The theoretical propositions help make the case for the foundations of their approach.

**Q4 Main Weakness:**

- The presentation of the work needs improvement, as it is hard to fully understand all the details of the work in the current format. Key metric definitions, interpretations of experiments should be included to make this paper more understandable for a reader.

- The main target of this paper is targeting inconsistent uncertainties that can arise from early exit neural networks. However, I believe we need to see more evidence on the scale of the problem arising from these inconsistent uncertainties in naive implementation of early exit neural networks.

**Q5 Detailed Comments To The Authors:**

- I understand the the predictions coming out of confidence intervals of early exit neural networks can be inconsistent. However, can you point out the scale of this problem for some benchmark datasets? For instance, let's say we're looking at the task of classification on ImageNet. Is there no way to account for the uncertainties coming out of these early exits that can help us understand whether that early exit prediction can be trusted or not, rather than just looking at confidence intervals? For instance look at [1].

- In section 2, when defining confidence intervals, $L_t$ and $R_t$ aren't defined. Although $R_t$ is defined later on, but it's also important to define the terms when you first introduce them.

- While defining AVCS in section 2, it's also important to denote what $\theta$ means in the context of defining the stochastic process$R_t(\theta)$ that's dependent only on $\mathbf{x}_t$. It's not clear what this stochastic process is.

- You should also describe why the posterior ratio is used to create valid AVCS as it's a key element of the paper. And also explain how you can extend this from the data generation example in section 2 to the discriminative classifier setting later on.

- Is proposition 1 a sufficient condition for AVCS?

- Is this approach flexible for any approximate Bayesian inference method? Let's say, I want to use SGHMC for classification setting. Can I extend this method for that?

- Why not use parallel AVCS for classification setting?

- Can you talk about how would you extract uncertainty metrics using your approach for downstream tasks, such as out of distribution detection, misclassification detection, etc.? For example, look at the tasks and uncertainty metrics referred in [2]

- Do we have an understanding of the "correctness" of this approach when it comes to experimental results? i.e. how accurate are the predictions coming out of this approach?

- The method of converting a standard pretrained early exit NN to DPN is not clear at all. Can you describe what the threshold $\tau_t$ means? And what's its connection with confidence sets here? Are confidence sets same as confidence intervals? What's the role of the activation function here? It's better to put an algorithm block for this.

- I think having a mathematically formulated definition of the marginal coverage metric is useful. Also, I saw that "efficiency" was noted as another metric, but didn't see it in the experiments.

- What does the blue shaded part mean in Figure 1 top row? What's the "intersection" line here? Is it the same as running intersection? The running intersection line should be consistent here right (i.e. non increasing)? Then why do we see zig zag pattern in the Figure 1 top row?

- How do we decide what's a reasonable $\alpha$ to choose?


References

[1]  Jia, Hong, Young D. Kwon, Dong Ma, Nhat Pham, Lorena Qendro, Tam Vu, and Cecilia Mascolo. "UR2M: Uncertainty and Resource-Aware Event Detection on Microcontrollers." arXiv preprint arXiv:2402.09264 (2024).

[2] Vadera, Meet, Jinyang Li, Adam Cobb, Brian Jalaian, Tarek Abdelzaher, and Benjamin Marlin. "URSABench: A system for comprehensive benchmarking of Bayesian deep neural network models and inference methods." Proceedings of Machine Learning and Systems 4 (2022): 217-237.

**Q9 Complying With Reviewing Instructions:**

Yes

---

> ### Author Rebuttal · Authors · 2024-04-02
>
> We thank the reviewer for their help in improving our work. We hope their questions will serve as a starting point for an interesting discussion during the rebuttal. Below we address concerns raised one by one.
>
> > The presentation of the work needs improvement, as it is hard to fully understand all the details of the work in the current format. Key metric definitions, interpretations of experiments
>
> We describe all three metrics used (efficiency, marginal coverage, consistency) in the paragraph Evaluation metrics in Section 6. Based on your comment, we will make sure to include a more precise definition of efficiency and marginal coverage in the camera-ready version. Moreover, we tried to make sure to describe all experimental results in detail in their respective sections (see Sections 6.1-6.3). If you have other concrete suggestions on what concretely is missing in the interpretation of experimental results, please let us know, and we will be happy to include those.
>
> > However, I believe we need to see more evidence on the scale of the problem arising from these inconsistent uncertainties in naive implementation of early exit neural networks.
>
> > I understand the the predictions coming out of confidence intervals of early exit neural networks can be inconsistent. However, can you point out the scale of this problem for some benchmark datasets? For instance, let's say we're looking at the task of classification on ImageNet
>
> We present empirical evidence for the issue of inconsistent prediction intervals in EENNs in all our experiments, see *Average Consistency* plots in top rows of Figures 1, 3 and 4. It can be seen that a naive implementation of prediction sets via either conformal prediction or Bayesian credible intervals/sets (dashed red or blue lines), yields inconsistent behaviour as demonstrated by the consistency metric being smaller than 1 for later exits. Moreover, we consider another naive baseline, namely that of taking a running intersection of either conformal or Bayesian credible intervals/sets (solid red or blue lines). While this running intersection resolves the issue of consistency, as denoted by the consistency metric equal to 1 across all exits, it leads to large decays in marginal coverage as shown in the *Marginal Coverage* plots of the respective figures (middle rows).
>
> Please, see the right-most column in Figure 4 which illustrates the inconsistent behaviour of naive baselines for the ImageNet dataset you mention.
>
> > Is there no way to account for the uncertainties coming out of these early exits that can help us understand whether that early exit prediction can be trusted or not, rather than just looking at confidence intervals?
>
> We agree that yielding prediction sets (classification) or intervals (regression) is only one way to account for the uncertainty in EENNs. Alternative ways include working with Bayesian predictive distribution directly [1] or relying on some more ad-hoc notion of uncertainty (e.g. top-softmax probability in the classification case). In our work, we focus exclusively on the uncertainty via prediction sets. We acknowledge that this has not been emphasised clearly enough in the current version. We will make sure to define the scope more clearly in the camera-ready, and we thank you for raising our awareness around this.
>
> > In section 2, when defining confidence intervals, $L\_t$ and $R\_t$ aren't defined. Although is defined later on, but it's also important to define the terms when you first introduce them.
>
>  $L\_t$ and $R\_t$ in Section 2, represent the left- and right-endpoint of the confidence interval $C\_t := (L\_t, R\_t)$, respectively. We see that the source of confusion might be due to our notation: $R_t$ is first used to denote the right-endpoint of the confidence interval and later on as the martingale process. We thank you for spotting this, we will fix this confusing notation in the camera-ready.
>
> > While defining AVCS in section 2, it's also important to denote what $\theta$ means in the context of defining the stochastic process $R_t(\theta)$ that's dependent only on $\mathbf{x}_t$. It's not clear what this stochastic process is.
>
> Thanks for raising this point, we see that we have to make a distinction between $\theta$ (a candidate parameter) and $\theta^*$ (the true parameter of the data-generating distribution, i.e. $\boldsymbol{x}_t \sim p(\mathbf{x} | \theta^*)$) more explicit. We also tried to help the reader by providing an example of prior-posterior ratio in the same paragraph. Will expand on this example further using the additional space in the camera-ready.

---

### Official Review · Reviewer_cBAH · 2024-03-22

**Q2-1 Originality-Novelty:** 3
**Q2-2 Correctness-Technical Quality:** 3
**Q2-5 Clarity Of Writing:** 3

**Q10 Ethical Concerns:**

No.

**Q1 Summary And Contributions:**

This paper proposes a method to construct anytime-valid confidence sequences for early-exit neural networks.  These confidence sequences can be applied for both regression as well as classification.

**Q2-3 Extent To Which Claims Are Supported By Evidence:**

3: Good: the main claims are supported by convincing evidence (in the form of adequate experimental evaluation, proofs, (pseudo-)code, references, assumptions).

**Q2-4 Reproducibility:**

3: Good: key resources (e.g. proofs, code, data) are available and key details (e.g. proofs, experimental setup) are sufficiently well-described for competent researchers to confidently reproduce the main results.

**Q3 Main Strengths:**

The paper is well written, demonstrating the intuition through the idealized construction as well as how to construct a practical estimator.  The relevant results are also proved theoretically and the experiments suggest that the results are promising.

**Q4 Main Weakness:**

None.

**Q5 Detailed Comments To The Authors:**

The figure captions are quite long and make it a bit difficult for the reader.  I think it might be easier if some of the information (such as the significance level) are included in the description of the experiment rather than in the figure caption.

It might also be helpful for the reader to mention that the two algorithms are located in the appendix.

In Section 2, why does R_t(θ^*) need to be discrete?  Doesn't Ville's inequality hold as long as it is a supermartingale?

**Q9 Complying With Reviewing Instructions:**

Yes

---

> ### Author Rebuttal · Authors · 2024-04-02
>
> We thank you for your review and we are encouraged that you find our work promising. Below we address your open comments.
>
> > The figure captions are quite long and make it a bit difficult for the reader.
>
> Noted, we will make sure to use the additional space in camera-ready to move parts of the figure’s captions into main text to improve readability.
>
> > It might also be helpful for the reader to mention that the two algorithms are located in the appendix.
>
> We do so already (see starting paragraphs in Sections 4 and 5) but will make sure to make the reference to the algorithms in Appendix E even more explicit.
>
> > In Section 2, why does R_t(θ^*) need to be discrete? Doesn't Ville's inequality hold as long as it is a supermartingale?
>
> Indeed, Ville’s inequality does not require discrete-time martingales [1]. However, since discrete time case is sufficient for the early-exit setting which is the main focus of our work (different exits correspond to a discrete sequence), we restrict our presentation of AVCS to discrete time for the ease of exposition.
>
> [1] https://www.stat.cmu.edu/~aramdas/martingales18/L13-Ville's.pdf

---

### Official Review · Reviewer_8ubD · 2024-03-26

**Q2-1 Originality-Novelty:** 2
**Q2-2 Correctness-Technical Quality:** 3
**Q2-5 Clarity Of Writing:** 4

**Q10 Ethical Concerns:**

No.

**Q1 Summary And Contributions:**

This work proposes an uncertainty estimation approach for the early-exit neural networks (EENNs) that aim to achieve consistency in the predicted uncertainty intervals in the sense that the dropped interval at previous time steps should not show up in intervals predicted by the following steps. It is achieved by applying a so-called anytime-valid confident sequences (AVCS) scheme to the EENNs. In the paper, the different treatments of regression and classification tasks when applying AVCS to EENNs are presented. Experimental evaluations on both tasks are further presented.

**Q2-3 Extent To Which Claims Are Supported By Evidence:**

2: Fair: the main claims are somewhat supported by evidence (but the experimental evaluation may be weak, or does not match entirely with the claims, important baselines may be missing, proofs contain important ideas but lack rigor, algorithmic details are only discussed superficially, references are imprecise, assumptions are not sufficiently motivated or explicated, etc.).

**Q2-4 Reproducibility:**

3: Good: key resources (e.g. proofs, code, data) are available and key details (e.g. proofs, experimental setup) are sufficiently well-described for competent researchers to confidently reproduce the main results.

**Q3 Main Strengths:**

This work is overall well-written and has provided sufficient background for the authors to understand the proposed approach. The proposed consistent EENNs via AVCS come with theoretical analysis.

**Q4 Main Weakness:**

- In the experiments, evaluations on uncertainty estimation and predictive performance are missing. Specifically, in Bayesian deep learning settings, to evaluate the uncertainty estimation accuracy, common metrics include the test log-likelihood and expected calibration error as in [1]; for predictive performance, results on RMSE for regression tasks and results for classification accuracy for classification tasks are expected but they are all missing in this work.
- The missing empirical evaluations as mentioned above lead to the question of whether enforcing the consistency using AVCS would harm the uncertainty estimation and predictive performance. Actually, in Figure 2 where the predictive confidence for both the proposed method EENN-AVCS and the baseline method EENN-Bayes are shown, the proposed EENN-AVCS has much worse performance than the baseline in the sense that for the out-of-distribution points, the variance is even much smaller than in-distribution ones.


References:
[1] Izmailov, Pavel, et al. "Subspace inference for Bayesian deep learning." Uncertainty in Artificial Intelligence. PMLR, 2020.

**Q5 Detailed Comments To The Authors:**

- Does the AVCS scheme harm the uncertainty estimation and preditive accuracy?
- For the regression task, it seems the proposed algorithm is for one-dimensional output. Does it generalize to multi-dimensional ones? I'm asking because in the algorithm, there is a step that requires to find the roots of equations. This might be costly for multi-dimensional outputs since it would require to find root in a potentially high-dimensional space.

**Q9 Complying With Reviewing Instructions:**

Yes

---

> ### Author Rebuttal · Authors · 2024-04-02
>
> We thank the reviewer for their efforts in reviewing our work. Below we address the raised concerns one by one.
>
> > In the experiments, evaluations on uncertainty estimation and predictive performance are missing. Specifically, in Bayesian deep learning settings, to evaluate the uncertainty estimation accuracy, common metrics include the test log-likelihood and expected calibration error […]
>
> We would point out that our EENN-AVCS yields a prediction set (classification) or interval (regression), i.e. a subset of $\mathcal{Y}$ which is meant to contain the ground-truth label $y^*$ with high probability. For models that yield prediction sets, the standard metrics to evaluate their performance is a combination of efficiency (i.e. the set size) and coverage (i.e. how often do they cover the ground-truth), see [1, 2, 3]. In our work we use precisely those two metrics, and we additionally introduce a consistency metric which becomes important when prediction sets are used in the early-exit setting.
>
> Note how this is different to Bayesian Deep Learning models that yield an entire predictive distribution and where metrics like log-likelihood and ECE are used to evaluate performance. For our Bayesian baselines (EENN-Bayes), we convert the predictive distributions into credible sets/intervals, since we are not aware of a way to compare prediction sets to predictive distributions. If we are wrong here, we would appreciate you pointing out a way to do so and we will incorporate such comparison in our work.
>
> We thank you for brining up this point as it made us aware that we have not made our focus on prediction sets/intervals clear enough. We should be more explicit that we go about uncertainty estimation not via predictive distributions (as in BDL case) but via prediction sets/intervals. We will do so in the camera-ready version.
>
> > Actually, in Figure 2 […] the proposed EENN-AVCS has much worse performance than the baseline in the sense that for the out-of-distribution points, the variance is even much smaller than in-distribution ones.
>
> This seems to be an unfortunate misunderstanding regarding Figure 2. Note that our EENN-AVCS collapses to empty prediction intervals in parts without training data (OOD), representing the maximal uncertainty, which is a desirable behaviour. We tried to make this very explicit by depicting the collapse to an empty interval with a red-dashed line in Figure 2 and also describing the OOD behaviour thoroughly (see the last two paragraphs in Section 6.1). Please let us know, if you have any other suggestions on how to make this more explicit, we would be happy to incorporate it.
>
> > Does the AVCS scheme harm the uncertainty estimation and preditive accuracy?
>
> As aforementioned, due to our model yielding prediction sets/intervals we evaluate the uncertainty estimation and predictive accuracy via a combination of efficiency and coverage. Moreover, one of our main contributions is to propose a new dimension of evaluation for prediction sets in the early-exit setting, namely their nestedness or consistency.
>
> We summarise results presented in Section 6 here: compared to the baselines we evaluated, our EENN-AVCS is the only approach that maintains consistency across exits while preserving high marginal coverage. However, it is true that this advantage comes at a cost in terms of efficiency - our prediction sets/intervals are notably larger, especially at the initial exits. We tried to be as transparent as possible about this limitation in our work. If you think we should highlight this aspect more, please let us know. While we recognize that enhancing efficiency is a vital direction for future research (as outlined in our work, see also Appendix B.1 for some preliminary results in this direction), we believe it is equally important to pursue this goal while keeping the consistency criterion in mind. It is our hope that our work provides a solid starting point in this regard.
>
> > For the regression task […] Does it generalize to multi-dimensional ones?
>
> Our focus is on regression with one-dimensional outputs. We will make sure to define the scope more precisely in the camera-ready. We are not aware, though, of prior work on prediction intervals for regression with multi-dimensional outputs. If you have any references here, we would appreciate those, and we will explore how we can extend our work to such settings.

---

### Meta-Review · Area_Chair_Ukxe · 2024-04-12

This work proposes anytime-valid confidence sequences for early-exit neural networks. The reviews for this work tended to be positive, with 2 accepts, 1 borderline accept, and 1 borderline reject. The reviewers praised the novelty of the work and found the theoretical analysis to be solid. The one reviewer who was slightly unfavorable on the paper found the empirical experiments to be lacking, though others praised them. All in all this work seems to make an important contribution.